cognition/computer graphics/computer modelling and simulation

racial bias, empathy, implicit association test, virtual reality, embodiment

**Author for correspondence:**
Mel Slater
e-mail: melslater@ub.edu

[†]Equal contributions.
[‡]Present address: Centre de la Imatge i la Tecnologia Multimèdia, Universitat Politècnica de Catalunya, Barcelona, Spain.

# Virtual body ownership and its consequences for implicit racial bias are dependent on social context

Domna Banakou[1,2], Alejandro Beacco[1,†],
Solène Neyret[1,†], Marta Blasco-Oliver[1], Sofia Seinfeld[3,‡]
and Mel Slater[1,2]

[1]Event Lab, Department of Clinical Psychology and Psychobiology, University of Barcelona, Passeig de la Vall d'Hebron 171, 08035 Barcelona, Spain
[2]Institute of Neurosciences of the University of Barcelona, Passeig de la Vall d'Hebron 171, 08035 Barcelona, Spain
[3]IDIBAPS, Rosselló, 149-153, 08036 Barcelona, Spain

DB, 0000-0002-0974-6971; AB, 0000-0001-8192-1431;
MB-O, 0000-0002-3812-1014; SS, 0000-0001-9649-0785;
MS, 0000-0002-6223-0050

When people hold implicit biases against a group they typically engage in discriminatory behaviour against group members. In the context of the implicit racial bias of 'White' against 'Black' people, it has been shown several times that implicit bias is reduced after a short exposure of embodiment in a dark-skinned body in virtual reality. Embodiment usually leads to the illusion of ownership over the virtual body, irrespective of its skin colour. Previous studies have been carried out in virtual scenarios that are affectively neutral or positive. Here, we show that when the scenario is affectively negative the illusion of body ownership of White participants over a White body is lessened, and implicit bias is higher for White participants in a Black virtual body. The study was carried out with 92 White female participants, in a between-groups design with two factors: BodyType (their virtual body was White or Black) and a surrounding Crowd was Negative, Neutral or Positive towards the participant. We argue that negative affect prevents the formation of new positive associations with Black and distress leads to disownership of the virtual body. Although virtual reality is often thought of as an 'empathy machine' our results suggest caution, that this may not be universally the case.

# 1. Introduction

Virtual Reality (VR) is often thought of as an 'empathy machine' [1], the idea being that embodiment of more privileged people in the bodies and situations of disadvantaged or discriminated-against groups using virtual reality would result in an increase in their empathy towards those groups. Here we show that the situation is more complex, and that while the empathy machine aim is laudable, the effect might be the opposite of that desired.

By 'embodiment' we mean that it is possible in VR to visually substitute a person's real body by a life-sized virtual one, seen from the person's own first-person perspective (1PP). When the person, wearing a wide field-of-view head-tracked head-mounted display, looks down towards themselves, they would see the virtual body coincident in space with, and substituting, their real body. Using real-time body tracking, when the person moves, the virtual body moves correspondingly and synchronously. They can also see their virtual body reflected in a virtual mirror. Seeing the virtual body from 1PP and the synchronous real-virtual body movement (visuomotor synchrony) are sufficient to induce a strong illusion of ownership and agency over the virtual body (for example [2]). The body ownership illusion is less likely to occur when the body is seen from third-person perspective (3PP) or when there is visuomotor asynchrony [3].

The virtual body ownership illusion is based on the paradigm of the rubber hand illusion (RHI) introduced by [4]. In the RHI set-up subjects can feel a rubber hand as their own when it is placed in an anatomically plausible position on a tabletop, the corresponding real hand is hidden, and the rubber and real hand are tapped and stroked synchronously temporally and spatially. The RHI is less likely to occur when the visuotactile stimulation is asynchronous. The illusion has also been shown to work with a virtual hand in immersive VR [5]. Visuotactile stimulation seen on a manikin body from 1PP in a head-mounted display was used in [6] to first demonstrate an illusion over the full body.

Body ownership illusions, whether in physical or virtual reality, have been attributed to a combination of bottom-up causes through visuotactile or visuomotor integration, and top-down influences, in particular the relationship of the fake body or body part to the body representation, which has to be consistent with top-down expectations [7,8]. For example, [9] showed that when the rubber hand is rotated to an anatomically implausible position then the ownership illusion does not occur. Hence although the bottom-up visuotactile or visuomotor synchrony is a necessary condition [9–11], it is not sufficient—top-down influences are important, especially the disposition of the surrogate body [12], and for example, continuity between body parts [13,14]. With respect to the full-body ownership illusion [15] investigated adults who were embodied in a child or adult body, which spoke with a child or adult voice. It was found that body ownership was reduced in the incongruent conditions—embodiment as a child speaking with an adult voice or as an adult with a child's voice. However, the congruent conditions reproduced the findings of [16] where body ownership of adults embodied in a child- or adult-shaped body were not different. In [15], the top-down expectation that the voice matches the body was important for body ownership.

A correlate of the body ownership illusion is that it has been shown that the *form* of the body can result in various changes in the participant. Yee & Bailenson [17] referred to this as the Proteus effect, showing, for example, that having more attractive virtual bodies changes peoples' proxemics behaviour, or that people negotiate differently (more or less aggressively) if their virtual body is taller or shorter. This can operate even at a physiological level—for example, body ownership over a transformed virtual body has also been found to influence pain threshold [18]. The size and shape of the body can influence spatial perception, for example, [19] showed that participants embodied as a Barbie doll perceived the world as larger. It was shown in [16] that embodiment of adult participants in the body of a 5-year-old child in VR resulted in their overestimation of object sizes by almost double that of a control group embodied in an adult-shaped body of the same size as the child, a result replicated in [15]. Moreover, participants in the child-embodiment group exhibited changes towards accepting implicit attitudes about themselves as being child-like.

With respect to implicit attitude changes, it was found in [20] that embodying White participants in a Black body for 12 min resulted in a reduction of their implicit bias against Black people, as measured by the implicit association test (IAT) [21], a result also shown for the RHI with White participants and a Black rubber hand [22]. The result for the virtual body has been replicated and found to last at least one week after the virtual reality exposure [23]. It was found in [24] that White participants in a Black virtual body will tend to unconsciously mimic the gestures and postures of a Black virtual partner more than a White one (or when embodied as White, imitate the White virtual partner more than a Black one), indicating that the chameleon effect [25], where mimicry indicates greater social rapport, also is influenced by body ownership. A review of the impact of embodiment on racial bias can be found in [26] and a neural network model that successfully simulates the IAT results is proposed in [27].

To date, almost all studies of body ownership take place in non-social and non-affective contexts. In the case of the RHI of course the experimenter is present, but nothing else socially significant is happening. One earlier study on racial bias had a result that was contrary to all later studies, since the implicit racial bias of White against Black increased after being embodied in a Black virtual body [28]. However, the context was not a socially neutral one, but a job interview, typically associated with stress and racial bias. In [24], the reduction in implicit racial bias was more likely to occur among those participants who liked their Black virtual partner. A study of the effects on bias against women found that implicit bias increased among men embodied as women compared with being embodied as men [29]. The authors argued that this was due to the stress of the task and the fact that the task itself (learning Tai Chi) was more associated with male rather than female stereotypes. These are clues that affect caused by the social situation may influence embodiment outcomes.

One further result of all these studies is that body ownership itself appears to be invariant to the form of the body. Adult participants are just as likely to report the illusion of body ownership over a child body or an adult body. White participants report the same level of body ownership over a Black body or a White body. In [30,31], participants were embodied alternately in a virtual body that bore a close resemblance to themselves based on a scan of their own body, and also in a much older looking body depicting Sigmund Freud. The levels of body ownership between these two self-representations were high and similar to each other.

We carried out an experiment where participants stood in a virtual street embodied in a White or Black body, with crowds of virtual people walking by. There were two factors: BodyType and Crowd. BodyType had two levels, embodied in a White or Black virtual body. Crowd had three levels: Negative—some passers-by acted negatively towards the participant such as standing and looking at them while frowning, and then turning away, Neutral—they did not act in any particular way but sometimes glanced at the participant if they passed nearby, Positive—some passers-by greeted the participant by acknowledging them in some way, such as nodding towards them with a pleasant facial expression. Based on our previous findings and those of other studies discussed above, we hypothesized that when White people are embodied in a Black virtual body, there would be a change in implicit racial bias that depends on the social situation depicted in the scenario. Specifically, although embodiment in Black can lead to a reduction in implicit racial bias in neutral and positive social situations as in [20,23] and discussed in [26,27], it will not necessarily lead to a reduction in implicit racial bias in the negative social context, but it could increase it as found in [28].

The results of this experiment challenge two of the findings discussed above. First, we show that body ownership itself can be influenced by the social context portrayed in the virtual setting and consequent affect. Second, we show that when White people are embodied in a Black body there is not necessarily a reduction in implicit racial bias, in fact it can increase. White people embodied in a White body in a social situation where negative affect is generated are likely to have less body ownership than when in a Black virtual body in a negative social situation.

# 2. Material and methods

## 2.1. Participants

Participants were recruited by advertisement and email around the campus of the University of Barcelona and nearby. All participants were Caucasian women. There is complete data on 92 female healthy participants (72 studying, 15 working, 5 both studying and working) aged 18–30 years (mean ± s.d. age 21.8 ± 3.14). Eleven additional participants were excluded from the analysis due to system failure during the experiment and due to withdrawal from the study after the first phase (i.e. incomplete data). All participants were compensated for their participation through a payment of €15 (Euros) (€5 after the end of the first phase, and the remaining €10 after completion of the second phase).

The design was between groups with 15 participants arbitrarily assigned to each cell except for the Neutral crowd which had 16 in each cell. Their education level was very similar across all cells of the design. The vast majority were students. They had low levels of computer science knowledge (programming), very little prior VR experience and hardly played computer games. Participants had no prior knowledge of the experiment. The experimental groups were comparable across several variables. Electronic supplementary material, table S1 shows for each group the total number of participants, means of ages, median values for participants' experience in VR and hours per week playing video games.

The experiment was approved by the Comissió Bioètica of Universitat de Barcelona and performed according to institutional ethics and international standards for the protection of human participants. Ethical considerations included written informed consent, right to withdraw (either during the experiment or later) and confidentiality. Exclusion criteria were epilepsy, use of medication, recent consumption of alcohol, mental health difficulties (requiring medication) and race other than Caucasian.

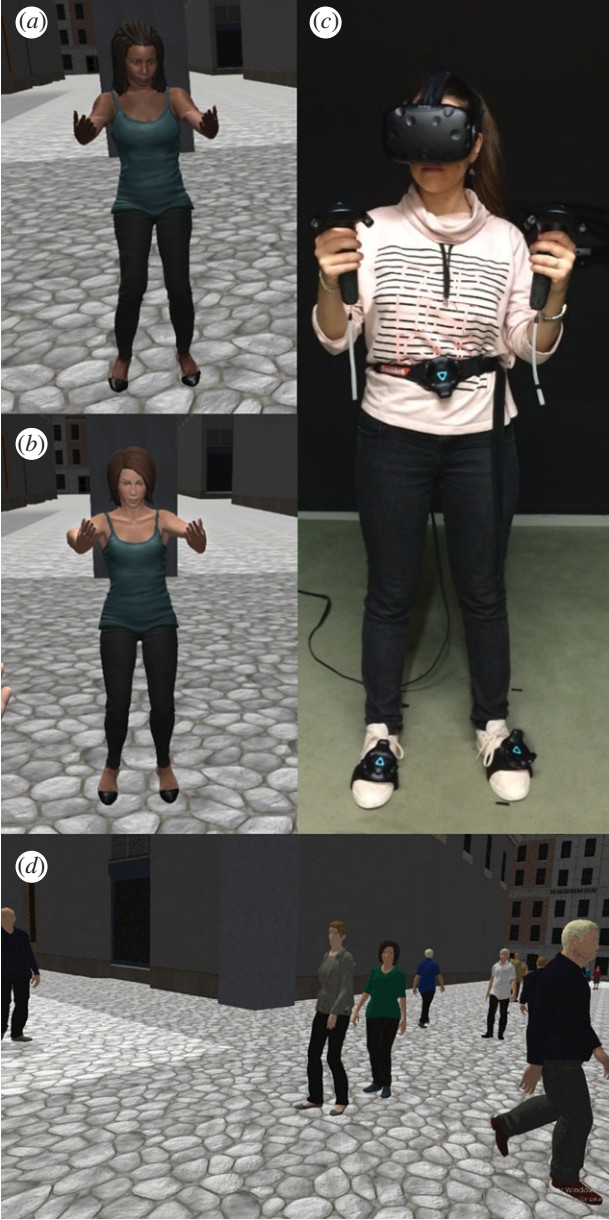

**Figure 1.** The scenario. (*a*) The participant embodied in a Black virtual body is in front of a mirror and carrying out embodiment exercises. (*b*) A participant in a White virtual body. (*c*) Showing the participant with the virtual reality equipment. (*d*) First-person view of the Negative crowd—note how two virtual women in the centre of the picture stare at the participant.

Following the completion of the last phase of the experiment, participants were debriefed about the purpose of the study.

## 2.2. Experimental set-up

Apart from the two experimental factors BodyType (White, Black) and Crowd (Negative, Neutral, Positive), a covariate that was considered was self-esteem as measured by the Rosenberg self-esteem scale [32] prior to the VR exposure. This was included because the theory developed in [27] suggests that participants in a Black virtual body are more likely to exhibit a reduction in racial bias the greater their self-esteem. The self-esteem scale ranges from 4 to 40, with higher values indicating greater self-esteem. In our sample, the scores were between 22 and 40, with mean ± s.d.: 31.3 ± 4.34. Esteem hardly varied in the cells of the design (electronic supplementary material, table S1): minimum 29.8 ± 0.98 s.e., to maximum 33.2 ± 1.13 s.e. The normal range for self-esteem is between 25 and 35. In our sample, there are 5 below 25 and 19 above 35. Esteem did not contribute at all to the results since there was an insufficient variation of esteem in the sample. Therefore, we do not consider it further.

**Table 1.** Questions on body ownership, agency and presence. The scores ranged from −3 (strongly disagree) to 3 (strongly agree).

| variable name | question |
|---|---|
| *mybody* | I felt that the virtual body I saw when looking down at myself was my own body. |
| *mirror* | I felt that the virtual body I saw when looking at myself in the mirror was my own body. |
| *twobodies* | I felt as if I had two bodies. |
| *notme* | I felt as if the virtual avatar was not me. |
| *features* | I felt that my virtual body resembled my own (real) body in terms of shape, skin tone, or other visual features. |
| *agency* | I felt that the movements of the virtual body were caused by my own movements. |
| *presence* | I felt I was in the middle of a crossroad with people walking by. |

The scenario is illustrated in figure 1, and highlights of the experiment can be seen in the electronic supplementary material, video S1.

## 2.3. Procedures

Participants visited the laboratory twice, once to complete some basic baseline measurements and questionnaires and then between 6 and 9 days later for their virtual reality exposure and collection of further post-exposure data. Participants attended the experiment at pre-arranged times.

Upon arriving on a first visit, they were given an information sheet to read and sign, and after they agreed to continue with the experiment, they signed a consent form. Next, they were seated in front of a desktop computer to complete a demographics questionnaire (i.e. age, occupation, VR and games experience etc.), and they were assessed on the Rosenberg self-esteem scale [32]. Then they completed the racial IAT [21] and the attitudes towards Blacks (ATB) scale [33], and the results were recorded (variables: *preIAT* and *preATB*). The whole procedure lasted approximately 20 min. Between 6 and 9 days later they returned for the VR exposure. More details of the IAT procedure are given in the electronic supplementary material, text S2, and the ATB in the electronic supplementary material, text S3.

In the VR laboratory, participants were fitted with the head-mounted display (HMD) and body tracking VIVE trackers (figure 1*c*). Upon entering the virtual environment, participants found themselves standing alone in the middle of pedestrian crossroads, surrounded by tall buildings. Their body was substituted by the Black or White virtual body, seen from 1PP. Their head and body movements were mapped in real-time to the virtual body. They saw their virtual body both by looking down directly towards it, and in a virtual mirror. A series of instructions were then given to them from a pre-recorded audio. First, they were asked to look around in all directions and describe what they saw, and then they were instructed to perform some body movement exercises to explore the capabilities and real-time motion of the virtual body, including movements of their arms, legs and feet. After this 5 min orientation phase, the scene faded out, and on returning, participants were in the same crossroads, but with the streets now busy with virtual human characters walking by in different directions. A virtual female character acting as the participant's 'friend' asked her to wait there while she went back to a café to look for her mobile phone.

During this time, which lasted 3 min, participants were left among the virtual crowd, which exhibited the different behaviours (Negative, Neutral or Positive) depending on the condition to which the participant had been assigned. When the virtual friend returned, she stated that she had found her phone, then the virtual scene faded out and the VR exposure finished. The HMD and trackers were then removed, and participants were asked to complete the racial IAT again (*postIAT*), along with two post-experience questionnaires (table 1 and electronic supplementary material, table S2), followed by the ATB questionnaire (*postATB*). The whole procedure lasted approximately 35 min. Two experimental operators (females) were present throughout the experiment. Further information is given in electronic supplementary material, video S1.

## 2.4. Virtual reality implementation

The experiment was conducted in a virtual reality laboratory (width: 3.5 m, length: 4.0 m—back wall to curtain—height: 2.5 m). Participants were fitted with an HTC VIVE head-mounted display (https://www.vive.com/us/product/vive-virtual-reality-system/) (figure 1*c*). This is a stereo HMD and has a nominal field-of-view of 100°, with a resolution of 2160 × 1200 pixels per eye displayed at 90 Hz.

Additional VIVE trackers were attached to participants' hands, waist and feet to support real-time whole-body tracking. The virtual environment was implemented on the Unity3D platform (unity.com). The animation-enabled models of the White and Black virtual bodies were created with Adobe Fuse academic version (www.adobe.com/es/products/fuse.html) and Mixamo (www.mixamo.com).

The virtual agents forming the crowd around the participant were created using Adobe Fuse and Mixamo. The crowd was composed of approximately 80 agents in total, with equal numbers of males and females. All agents were designed to be White. They walked past the participant and along the virtual crossroads following random paths and directions, keeping a minimum distance of 2 m from the participant in all conditions. The virtual agents were designed to exhibit three different types of social behaviour towards the participant, which reflected changes in three specific parameters. These included facial expressions, eye contact with the participant, avoidance of the participant (i.e. changing direction when they saw the participant from a distance) and acknowledgement of the participant (i.e. saluting the participant by nodding when they passed near her). In the *Positive* condition, the virtual agents were designed to have happy facial expressions (e.g. smiling), occasionally looking (40% of the total duration of the experimental phase) and nodding towards the participant, and they never changed direction to move away from her. In the *Negative* condition, the crowd was designed to have negative facial expressions (exhibiting anger and frustration), their gaze was frequently directed towards the participant (70% of the total duration of the experimental phase), while they were also programmed to change direction and walk away from the participant (70% of the total duration of the experimental phase) without ever nodding. Finally, in the *Neutral* condition, the crowd had neutral facial expressions; they never looked at the participant directly except by chance, or nodded, and never changed direction. The different parameters were decided following pilot experimentation with the aim to evoke in participants feelings of acceptance and positive mood (*Positive* condition), rejection and discomfort (*Negative* condition), or no specific emotions (*Neutral* condition). Further information about the implementation of the crowd is given in electronic supplementary material, text S1.

## 2.5. Response variables

The implicit association test for racial bias (IAT) was administered between 6 and 9 days before the VR exposure (leading to the score *preIAT*) and immediately after the exposure (*postIAT*). The IAT score ranges from −2 to 2, where higher scores indicate greater implicit bias of White against Black (electronic supplementary material, text S2). We also measured explicit racial bias with the 'attitude to Blacks' scale [33] in the same sessions, leading to pre- and post-scores *preATB* and *postATB* (electronic supplementary material, text S3). The ATB questionnaire used was slightly modified since one of the 20 questions could not apply in Spain. The score is between 19 and 95 and higher scores correspond to less explicit bias. Body ownership, agency and presence were assessed by the questionnaire items shown in table 1.

A questionnaire (electronic supplementary material, table S2) concerned with how participants evaluated the crowd was administered after the VR exposure, in order to check that their response was in accord with the goals of the experiment (i.e. that the *Negative* condition would result in negative affect).

## 2.6. Chronbach's alpha for body ownership questions

A principal components factor analysis on the variables *mybody*, *mirror* and *notme* results in one factor accounting for 69% of the total variance. This has positive factor loadings for *mybody* (0.80) and *mirror* (0.88) and a negative factor loading for *notme* (−0.82). The factor scores give a new variable $0.38 \cdot mybody + 0.42 \cdot mirror - 0.39 \cdot notme$, which is almost the same as $mybody + mirror - notme$. In line with the use of Cronbach's alpha which assumes summative scores, we therefore construct a new variable *yown* as $(mybody + mirror - notme)/3$, so that $yown = 0$ when $mybody = mirror = notme = 0$, $yown = 3$ when $mybody = mirror = 3$ and $notme = -3$, and $yown = -3$ when $mybody = mirror = -3$ and $notme = 3$. Chronbach's alpha for *mybody*, *mirror* and *notme* is 0.78.

## 2.7. Statistical model

For each of the response variables (*postIAT*, *postATB*, *yown*), the statistical model is similar to a standard two-way ANOVA with one covariate:

$$\text{BodyType} + \text{Crowd} + \text{BodyType} \times \text{Crowd} \, [+ \, pre].$$

The covariate 'pre' refers to *preIAT* in the case of *postIAT* and *preATB* in the case of *postATB* and does not apply to *yown*.

A Bayesian model is used that includes all response variables simultaneously. This obviates the need for ad hoc control of the significance level in multiple comparisons, since all inferences are based on the joint posterior distribution of all the model parameters. Each likelihood (distribution of the response variable conditional on the parameters) is a Student $t$ distribution. This was used to flexibly incorporate outliers, since with low degrees of freedom this distribution has much wider effective support than the normal, yet for high degrees of freedom is approximately the same as the normal. The degrees of freedom of the distributions (and the scale factors) are also parameters with prior distributions.

For notational convenience, we use $b_i$ to denote the BodyType of the $i$th individual in the sample where $b_i = 1, 2$ for White, Black respectively. Similarly, we use $c_i$ to denote Crowd, where $c_i = 1, 2, 3$ for Negative, Neutral and Positive, respectively. $y_{iat,i}$, $y_{atb,i}$, $y_{own,i}$ denote the response variables (i.e. *postIAT*, *postATB*, *yown*) for the $i$th individual.

The notation $\text{Student\_}t(\nu, \mu, \sigma)$ indicates a Student $t$ distribution with degrees of freedom $\nu > 1$, mean $\mu$, and scale parameter $\sigma > 0$.

Then the likelihood for the IAT response is

$$y_{iat,i} \sim \text{Student\_}t(\nu_{iat}, \ \mu'_{iat}[b_i, c_i] + \beta_{iat} \cdot y_{preiat,i}, \ \sigma_{iat})$$
$$i = 1, 2, \ldots, n. \tag{2.1}$$

Here, $\nu_{iat} > 1$ is the degrees of freedom parameter of the Student $t$ distribution, $\mu'_{iat}[b_i, c_i]$ is the mean of the distribution for the BodyType and Crowd specified by $b_i$, $c_i$ for the $i$th individual, apart from the influence of *preIAT*. Hence $\mu'_{iat}$ is a $2 \times 3$ array of parameters, corresponding to the experimental design. $\sigma_{iat} > 0$ is the scale parameter for Student $t$. The variable $y_{preiat,i}$ is the *preIAT* covariate with coefficient $\beta_{iat}$.

Note that $\mu'_{iat}[i, j]$ is the effect of the BodyType $i$ and Crowd $j$. The usual two-way ANOVA model is of the form

$$\mu'_{iat}[i, j] = \mu + \alpha[i] + \beta[j] + \gamma[i, j],$$

where

$$\sum_{i=1}^{2} \alpha[i] = 0, \ \sum_{j=1}^{3} \beta[j] = 0 \ \sum_{i=1}^{2} \gamma[i, j] = 0 \ \sum_{j=1}^{3} \gamma[i, j] = 0. \tag{2.2}$$

Here, $\alpha[i]$ is the effect of the $i$th BodyType ($i = 1, 2$ corresponding to White, Black), $\beta[j]$ is the effect of the $j$th Crowd condition ($j = 1, 2, 3$ corresponding to Negative, Neutral, Positive), and $\gamma[i, j]$ is the interaction term.

In order to obtain the sole effect of BodyType and Crowd removing the influence of *preIAT*, we should consider $\mu'_{iat}[i, j] - \mu$. However, from equation (2.2), $\mu$ is just the mean over $i$ and $j$ of $\mu_{iat}[i, j]$. Therefore from the posterior distributions of $\mu'_{iat}[i, j]$, we subtract the mean so that the final parameters of interest are

$$\mu_{iat}[i, j] = \ \mu'_{iat}[i, j] - \ \text{mean}(\mu'_{iat}). \tag{2.3}$$

The likelihoods for the other three response variables follow similarly:

$$y_{atb,i} \sim \text{Student\_}t(\nu_{atb}, \ \mu'_{atb}[b_i, c_i] + \beta_{atb} \cdot y_{preatb,i}, \ \sigma_{atb}),$$
$$y_{own,i} \sim \text{Student\_}t(\nu_{own}, \ \mu_{own}[b_i, c_i], \ \sigma_{own}) \tag{2.4}$$

and

$$\mu_{atb}[i, j] = \mu'_{atb}[i, j] - \text{mean}(\mu'_{atb}). \tag{2.5}$$

All of the mean parameters $\mu'$ and the two coefficients $\beta_{iat}$ and $\beta_{atb}$ are assigned weakly informative [34] normal prior distributions with mean 0 and standard deviation 10, so that all prior 95% credible intervals are $-20$ to $20$. The degrees of freedom $\nu$ and the scale parameters $\sigma$ are assigned weakly informative Gamma distributions with shape parameter 2 and rate parameter 0.1. This has equal tails 95% credible interval from 2.4 to 55.7.

The interest focuses mainly on the posterior distributions of the means $\mu_{iat}, \mu_{atb}, \mu_{own}$.

**Table 2.** Summaries of the posterior distributions of all parameters, showing the means and standard deviations of the posterior distributions, and the 95% credible interval. Prob > 0 are the posterior probabilities of the parameters being greater than 0.

| | mean | s.d. | 2.5% | 97.5% | Prob > 0 |
|---|---|---|---|---|---|
| *postIAT* | | | | | |
| White, negative | 0.04 | 0.09 | −0.14 | 0.21 | 0.664 |
| White, neutral | 0.11 | 0.08 | −0.05 | 0.27 | 0.918 |
| White, positive | 0.03 | 0.08 | −0.14 | 0.20 | 0.643 |
| Black, negative | 0.11 | 0.09 | −0.07 | 0.29 | 0.896 |
| Black, neutral | −0.09 | 0.08 | −0.25 | 0.07 | 0.136 |
| Black, positive | −0.20 | 0.09 | −0.37 | −0.03 | 0.013 |
| $\beta_{iat}$ | 0.37 | 0.10 | 0.17 | 0.57 | |
| $\sigma_{iat}$ | 0.34 | 0.03 | 0.28 | 0.40 | |
| $v_{iat}$ | 22.76 | 14.04 | 5.46 | 58.43 | |
| *postATB* | | | | | |
| White, negative | 0.89 | 0.86 | −0.77 | 2.52 | 0.850 |
| White, neutral | 0.27 | 0.87 | −1.46 | 1.99 | 0.626 |
| White, positive | 0.64 | 0.83 | −1.00 | 2.25 | 0.782 |
| Black, negative | −0.75 | 0.88 | −2.45 | 1.00 | 0.195 |
| Black, neutral | 0.25 | 0.81 | −1.36 | 1.85 | 0.631 |
| Black, positive | −1.30 | 0.88 | −3.02 | 0.47 | 0.067 |
| $\beta_{atb}$ | 0.98 | 0.04 | 0.91 | 1.05 | |
| $\sigma_{atb}$ | 3.50 | 0.30 | 2.94 | 4.12 | |
| $v_{atb}$ | 25.62 | 14.63 | 6.68 | 62.78 | |
| *yown* | | | | | |
| White, negative | 0.31 | 0.33 | −0.35 | 0.96 | 0.827 |
| White, neutral | 1.59 | 0.30 | 1.01 | 2.17 | 1.000 |
| White, positive | 1.23 | 0.32 | 0.58 | 1.86 | 1.000 |
| Black, negative | 1.55 | 0.31 | 0.94 | 2.14 | 1.000 |
| Black, neutral | 1.30 | 0.33 | 0.67 | 1.94 | 1.000 |
| Black, positive | 1.15 | 0.33 | 0.48 | 1.79 | 1.000 |
| $\sigma_{own}$ | 1.16 | 0.12 | 0.93 | 1.40 | |
| $v_{own}$ | 19.14 | 13.08 | 4.17 | 53.14 | |

The Stan system (https://mc-stan.org) [35] was used for the Bayesian analysis through the RStudio interface (https://mc-stan.org/users/interfaces/rstan). Four thousand iterations were used with four chains. Stata 16 (www.stata.com) was used for the factor analyses and all box plots and bar charts. The R library 'bayesplot' (https://mc-stan.org/bayesplot/) [36] was used to plot posterior distributions and 'loo' (https://mc-stan.org/loo/) used for the leave-one-out analysis (see below).

# 3. Results

Table 2 shows summaries of the posterior distributions of all parameters shown in equations (2.1) and (2.4). We refer to this table in the remainder of this section. Notice how all of the credible intervals for the $\mu$ and $\beta$ parameters substantially narrow from the prior range of −20 to 20. The credible intervals for the degrees of freedom ($v$) seem to be wide. However, for the prior distribution Gamma(shape = 2, rate = 0.1), the probability of a value greater than 30 is 0.8. For the posteriors $P(v_{iat} > 30 \,|\, \text{data}) = 0.24$, $P(v_{atb} > 30 \,|\, \text{data}) = 0.31$, and $P(v_{\text{own}} > 30 \,|\, \text{data}) = 0.17$. The value 30 is important because with degrees of freedom 30 or higher the Student $t$ distribution closely approximates the normal. This supports the use of the Student $t$ rather than the normal for these data.

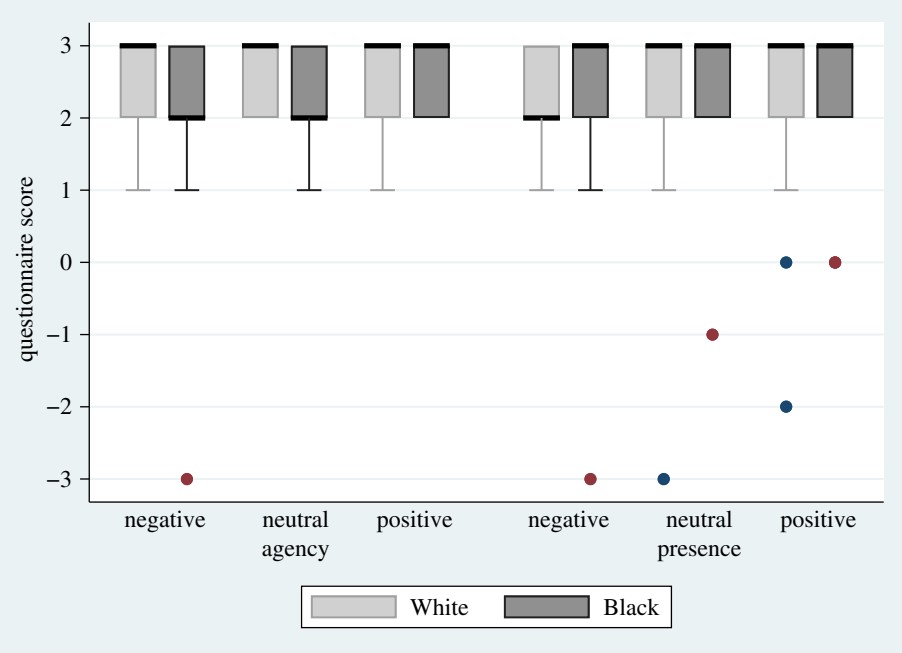

**Figure 2.** Box plots for the questionnaire scores on agency and presence. Box plots showing the medians and interquartile ranges (IQR) for presence and agency. The medians are the thick horizontal bars and the IQR the boxes. The whiskers extend from max (min score, lower quartile − 1.5*IQR) to min (max score, upper quartile + 1.5*IQR). The questions are shown in table 1.

We now consider each of the response variables.

## 3.1. Responses to the crowd

For the experimental study to be valid, participants should have experienced the Negative condition as an affectively negative experience compared with the Neutral and Positive conditions. A post-exposure questionnaire on responses to the crowd is shown in electronic supplementary material, table S2, and analysis is given in detail in electronic supplementary material, text S4. A principal components factor analysis over the questionnaire resulted in two factors accounting for 69% of the variance. The first factor was dominated by the crowd avoiding and keeping a distance from the participant, and the second factor was dominated by the negative gaze of the crowd towards the participant. Both factors were such that higher scores indicated greater negative affect. A Bayesian ANOVA model on the means of these two (uncorrelated) factors shows that for the first factor the Negative condition had a posterior probability of 0.956 of being greater than the Neutral and Positive, and a probability of 0.850 for the second factor. Full details are given in electronic supplementary material, text S4 and figures S1–S4.

## 3.2. Agency and presence

The results of the questions in table 1 are shown in figure 2. Agency and presence had uniformly high scores, with almost all medians 3, except for two medians at 2, and all interquartile ranges (IQR) between 2 and 3.

## 3.3. Body ownership

Figure 3*a* shows that in the Neutral and Positive crowd conditions all the medians for *mybody* and *mirror* (table 1) are at least 1, and all lower quartiles are at least 0, whereas for *notme* all medians are at most −1 and all upper quartiles but one are at most 0. The (White, Negative) for *mirror* is an exception to this where the median is 0 and the IQR spans from −2 to 2. Figure 3*b* shows the bar charts for *yown*. All means are at least 1 except for (White, Negative) which has mean 0.31 (s.e. 0.31).

Figure 4 shows the posterior distributions for the $\mu_{own}$. From the joint posterior distribution, the probability that (White, Negative) is less than all of the others is 0.922.

The Neutral and Positive conditions replicate previous work [20,23]. We should find that there is little or no difference in body ownership between White and Black embodiment in these cases. In the case of

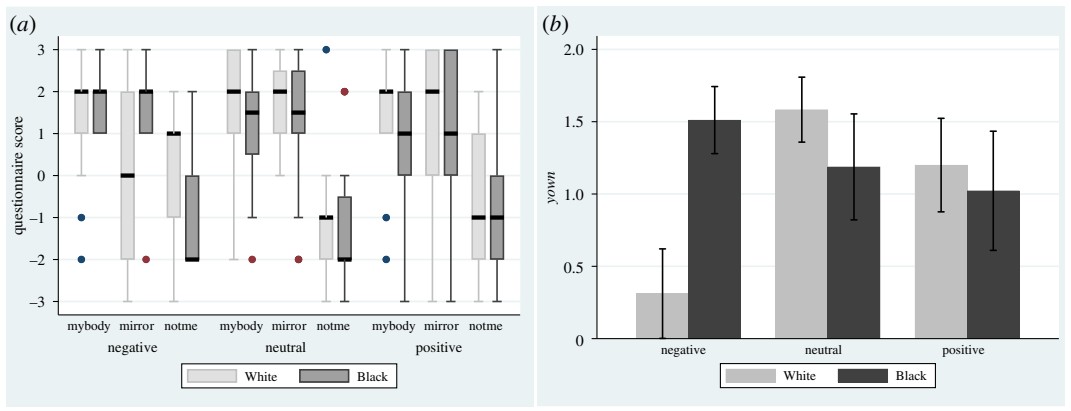

**Figure 3.** Questionnaire scores on ownership (*a*) box plots showing the medians and interquartile ranges. (*b*) Bar charts showing the means and standard errors for the summative score *yown*.

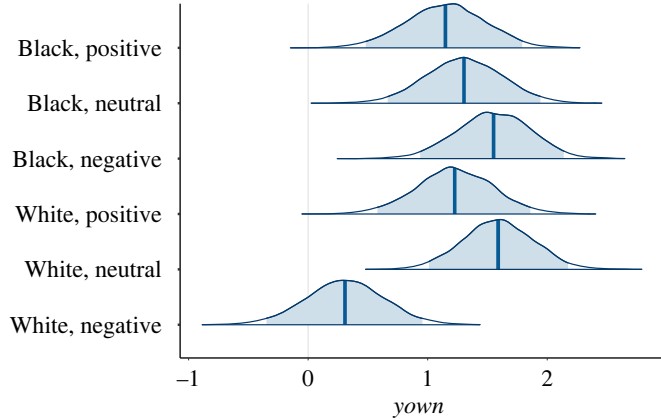

**Figure 4.** Posterior probability distributions for $\mu_{own}$. The shaded areas show the 95% credible intervals.

*yown*, the probability of the mean of (White, Neutral) being greater than (Black, Neutral) is 0.744. There is no evidence that (White, Neutral) is different from any other condition except (White, Negative).

The overall result from the analysis is that there is strong body ownership for (Black, Negative), and similar levels for all other conditions except for (White, Negative) which has a high probability of being less than all the others.

## 3.4. Explicit bias (ATB)

The level of explicit bias was low both before and after the exposure. Recalling that higher ATB values mean lesser bias, the *preATB* mean ± s.d. is 86.2 ± 6.39 (range from 66 to 95), and the *postATB* values are 85.0 ± 7.00 (range from 65 to 95).

From table 2, the coefficient of *preATB* has posterior distribution with mean of 0.98 (95% credible interval 0.91 to 1.05) (the correlation between *preATB* and *postATB* is $r = 0.86$). *postATB* and *preATB* have a strong linear relationship with approximately slope 1. Hence, as can anyway be seen from the means, standard deviations and ranges, ATB was unaffected by the experimental factors, as shown in figure 5. Figure 5 does suggest, however, that (Black, Positive) is less than (White, Negative), and the probability of this is 0.948. However, the mean ± s.d. of the distribution of the difference is −2.19 ± 1.36. Given the range of ATB, this is a very small effect size and can be ignored.

## 3.5. Implicit bias (IAT)

The mean and standard deviation of *preIAT* are 0.50 ± 0.40 ranging from −0.51 to 1.36; 12% are in the range 0.15 but less than 0.35 indicating a slight bias for White, 26% are in the range 0.35 but less than 0.65 indicating a moderate preference for White, and 39% are at least 0.65 indicating a strong preference for White. For *postIAT*, the corresponding values are 0.39 ± 0.39 with range −0.78 to 1.33,

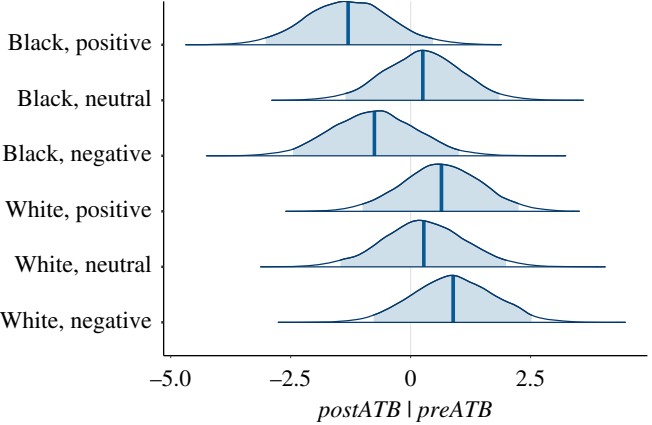

**Figure 5.** Posterior distributions of $\mu_{atb}$ conditional on the covariate *preATB*. The shaded areas show the 95% credible intervals.

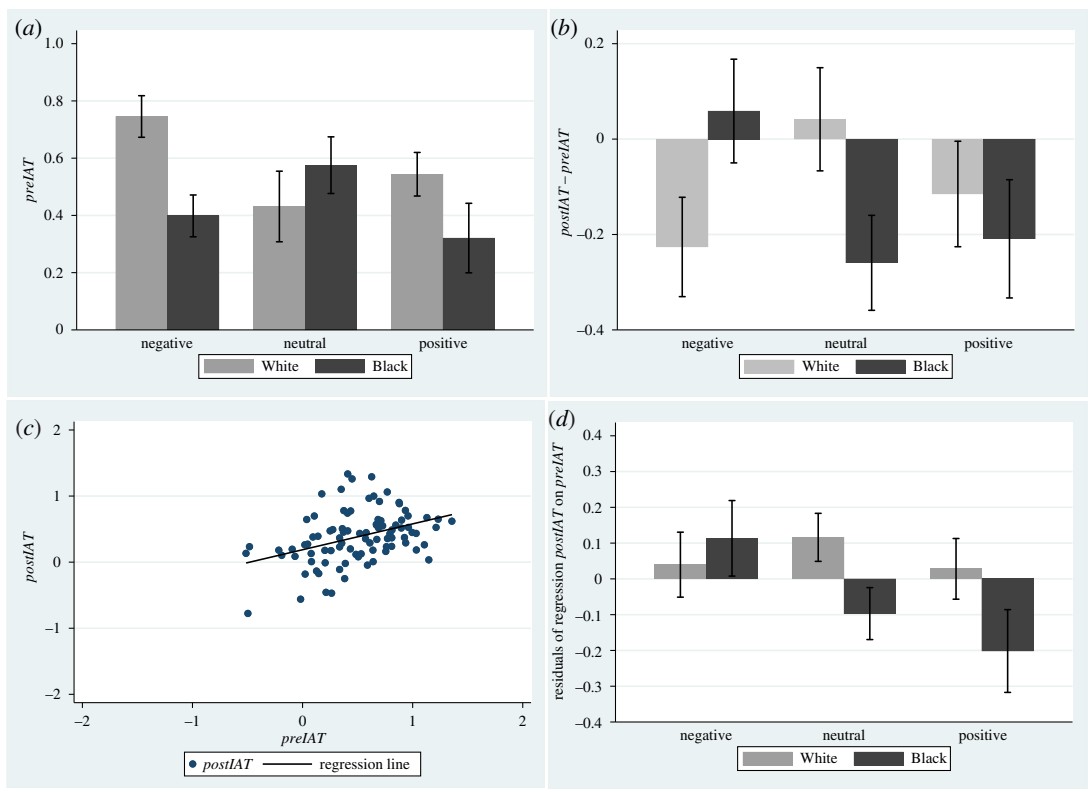

**Figure 6.** Implicit bias as measured by the IAT. (*a*) Bar chart showing means and standard errors for *preIAT*. (*b*) Bar chart for *postIAT – preIAT*. (*c*) Scatter diagram of *postIAT* on *preIAT*. (*d*) Bar chart of the residual errors of the regression of *postIAT* on *preIAT*.

and 21% show a slight bias, 35% show a moderate bias, and 21% are strong. The classifications of slight, moderate and strong are from [21,37,38].

Figure 6*a* shows the means and s.e.s for *preIAT*. As can be seen, by chance, the *preIAT* was greater for the (White, Negative) condition than all the others. (As an indicator of the strength of the difference, the Wilcoxon Rank Sum statistic is z = −2.72, p = 0.0057). The differences *postIAT* − *preIAT* are shown in figure 6*b*. However, these are misleading, since figure 6*c* shows that although the relationship between *postIAT* and *preIAT* is linear, the regression line is far from a 45° line, indicating that the effect on *postIAT* was influenced by *preIAT* differently for different values of *preIAT*. Hence simply considering differences *postIAT*–*preIAT* would not be appropriate. If $postIAT = a + b \cdot preIAT$ then $postIAT - preIAT = a + (b - 1) \cdot preIAT$. Since b < 1 greater values of *preIAT* will magnify the differences.

Hence it is important to use *postIAT* as the response variable and *preIAT* as a covariate, as in equation (2.1). The same method was used in [23]. This is supported by the results in table 2. The relationship between *postIAT* and *preIAT* is strongly linear. The posterior probability of the coefficient of *preIAT* (the

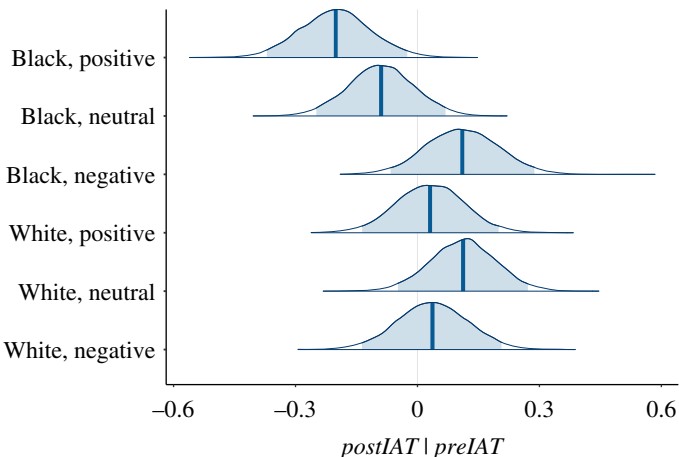

**Figure 7.** Posterior distributions of $\mu_{iat}$ conditional on the covariate *preIAT*. The shaded areas show the 95% credible intervals.

slope $\beta_{iat}$) being less than 1 is 1.000, with 95% credible interval 0.17 to 0.57, with the mean of the distribution being 0.37 (s.d. 0.10). This corresponds to figure 6c. Hence, other things being equal, there is a general reduction in IAT post-exposure.

Figure 6d shows the bar chart for *postIAT* by the conditions after removing the linear influence of *preIAT*: i.e. the residual errors of the regression line of *postIAT* on *preIAT*. Hence this illustrates how the variation in *postIAT* unexplained by *preIAT* may be influenced by the experimental factors. The Neutral condition replicates the results of previous experiments, with Black embodiment associated with a reduction in IAT (less bias) and White embodiment an increase. The lower IAT with Black embodiment seems to be exacerbated in the Positive condition. However, there is a difference in the Negative condition with Black embodiment associated with an increase in IAT.

Figure 7 shows the posterior distributions of the $\mu_{iat}$. Note that these distributions correspond well with figure 6d. From the joint posterior distributions, $P(\mu_{iat}[\text{White, Neutral}] > \mu_{iat}[\text{Black, Neutral}] \mid \text{data}) = 0.947$. The mean $\mu_{iat}$ for (Black, Positive) is less than all other conditions except (Black, Neutral) with each individual pairwise comparison with probability at least 0.957. The probability that it is simultaneously less than all others except (Black, Neutral) is 0.918.

The mean for (Black, Negative) is greater than (Black, Neutral) with probability 0.935, and greater than (Black, Positive) with probability 0.988. It is, though, approximately the same as the White embodiment conditions.

For example, the posterior distribution of the difference (White, Neutral) – (Black, Negative) ($\mu_{iat}[1, 2] - \mu_{iat}[2, 1]$) is symmetric and bell-shaped with mean 0. The 95% credible interval for this difference is −0.26 to 0.26 indicating no difference between these conditions.

## 3.6. Goodness of fit and model comparisons

The model converged with all Rhat values being 1, where Rhat is a measure of how much the different chains of simulation converged. Predicted posterior distributions of each of the response variables were computed, i.e. new data were simulated based on the model. These result in distributions for each of the individuals, and the means of those distributions can be used as point estimates for the predicted values. Correlations with the observed values were found and are shown in table 3. All of the correlations ($r$) are medium to high when considered as effect sizes. The strength of fit is also indicated by 95% confidence intervals for each $r$.

Additionally, to test the predictive ability of the model we used the 'leave one out' (loo) method [39] which leaves out each data point in turn, fits the model with the remaining data and estimates the one left out. This provides an 'out-of-sample' estimate of fit (i.e. each 'left-out' data point is not used to estimate the model that then predicts it). This results in an information-based statistic (expected log pointwise predictive density; ELPD) that showed no problems with the model: the so-called 'pareto $k$ estimates' were all less than 0.5 indicating no convergence problems for any individual, and also no overfitting was indicated (comparing the estimated effective number of parameters with the actual number used). The results are in table 4.

**Table 3.** Pearson correlations between observed and simulated values from the model. The simulated values are the means of the predicted posterior distributions.

| | Pearson Correlation | 95% confidence interval for $r$ | |
|---|---|---|---|
| variable | $r$ ($n = 92$) | 2.5% | 97.5% |
| postIAT | 0.49 | 0.317 | 0.631 |
| postATB | 0.87 | 0.805 | 0.910 |
| yown | 0.32 | 0.123 | 0.492 |

**Table 4.** The 'leave one out' (loo) method results for the model with *postIAT* as the response compared with *dIAT = postIAT − preIAT* as the response. All pareto *k* values are less than 0.5 indicating good convergence and predictive value. The p_loo parameter is an estimate of the effective number of parameters closely conforming to the actual number of parameters used, indicating no overfitting. The pairwise difference is between the ELPD of *postIAT* model minus the *dIAT* model.

| response variable | ELPD | s.e. | p_loo | s.e. |
|---|---|---|---|---|
| postIAT | −40.1 | 7.1 | 7.7 | 1.0 |
| dIAT | −57.0 | 6.5 | 6.7 | 0.8 |
| Pairwise difference | 16.8 | 5.0 | | |

The same method was used to compare the goodness of fit of the two models (i) using *postIAT* as the response variable with *preIAT* as a covariate, as described earlier, and (ii) using *dIAT = postIAT − preIAT* as the response variable. The ELPD was computed for each model and the pairwise difference between them and the standard error of the difference found. If the *dIAT* model were used instead the ELPD decreases by 16.8 with s.e. 5.0. Hence using the slightly more complex model (one parameter extra) is worthwhile, since it improves the predictive ability.

## 4. Discussion

There are two main findings from the experiment. The first is that body ownership may depend on the affective social situation depicted in the scenario. The second is that the change in implicit racial bias among White people embodied in a Black virtual body may also be influenced by the social situation depicted in the scenario. In particular, an affectively negative situation increases implicit bias compared with a neutral or positive situation, where previous findings have indicated a decrease in bias for neutral and positive.

First, we discuss the finding that body ownership was impacted by the affective response to the crowd. Illusory ownership among White participants over a Black or White rubber hand [22,40,41] and a Black or White virtual body [20,23,24] has been replicated a number of times, and no differences were found between the subjective illusion of ownership between the virtual Black and White conditions. The same was found in [42] except that the time to onset of the illusion was longer for the Black rubber hand than the White. The Neutral and Positive conditions of our experiment provide a further replication of these findings. The fact that the illusion is apparently impervious to skin colour changes leads to the idea that it is a bottom-up phenomenon caused by 1PP and multisensory integration of vision and touch in the case of the RHI, and vision and motor in the case of the virtual body, as in the present case.

It is known that participants respond physiologically to threats against the surrogate body (e.g. increased skin-conductance responses and greater heart-rate deceleration with respect to the rubber arm, virtual arm or whole surrogate body) [3,6,43], and also with corresponding brain activation, such as Mu-rhythm event related desynchronization (ERD) in the motor cortex and observed readiness potential (C3–C4) negativity, or increased activity in medial wall motor areas that reflects an impulse for withdrawal [44,45]. It has been argued that when the rubber hand is incorporated into the body representation it becomes worth defending, so that this should cause an increase in the ownership illusion [46]. However, here we consider not a physical threat to the virtual body but social discomfort. The response to the social threat with respect to body ownership in our case only makes

sense if there is implicit racial bias. No effect of the type of crowd on body ownership should have been found if ownership only requires appropriate bottom-up stimulation (in this case 1PP of the virtual body and visuomotor synchrony) and top-down with respect to the body shape (being human-like and with plausible movements—in this case the participant's own movements). However, White participants in a White-skinned body being treated negatively reported lower body ownership than would be expected from previous studies and in comparison with the other conditions in this experiment. It is as if their thoughts were 'This can't be happening, so this is not me'. Ataria [47] has argued that there is a direct connection between body ownership and traumatic events: '… as the traumatic experience becomes harsher the sense of ownership decreases: the sense that this is my body grows weaker'. It becomes a way of dissociating the self from unfolding traumatic events as a defence mechanism. While the negative crowd was certainly not traumatic, it was perceived as unpleasant (electronic supplementary material, text S4), and withdrawing from the situation through a reduction of body ownership would be a strategy to lessen the negative affect.

These results point to the conclusion that whereas bottom-up stimuli and top-down in relation to aspects of the surrogate body, and consistency between these two, are required for illusory body ownership, the social situation in which the embodiment takes place is also important. A social situation that does not match the expectations of participants with respect to the particular type of body, causing negative affect, is likely to reduce the likelihood of the body ownership illusion, as happened in the Negative condition as observed in the present study.

From a neural perspective, these results are in line with the findings of [48] that investigated the effect of race on the neural network associated with viewing affective body images among White participants. In an fMRI study, they found that the responses to affective body images influenced brain responses differentially between Black and White body images. In particular, when the body images were Black the responses were driven by positive emotions, but when the body images were White the responses were driven by negative emotions. This was among a population that had very little interaction with Black populations, as is the case with the sample in our study.

Moving now to the second main finding, the issue of implicit racial bias of White against Black, it has been shown multiple times that illusory ownership over a Black rubber hand [22,40–42] or a whole Black virtual body [20,24] leads to its reduction when White people are embodied as Black, an effect that may be sustained for at least one week after the exposure [23]. In the present work, we have found the (Black, Neutral) condition to be associated with lower IAT, and with a greater IAT for those in the (White, Neutral) condition, following the same pattern as in [20,23]. Additionally, the Positive condition is associated with a reduction in bias for Black.

Bias is related to bodily resonance, a function of how much another person or members of a group have features that are similar to the self-image. It was found in [49] that in the context of the RHI, the features of the rubber hand did not influence the extent to which the illusion was attained, but those who did experience the illusion would, afterwards, be more likely to report a similarity between their own hand and the rubber hand, even though objectively this was not the case. Following from this, the theory is that embodiment in Black leads to greater bodily resonance with the other-race group. The IAT specifically measures pre-existing cognitive associations between groups (represented by faces) and evaluative categories (the words). These associations are likely to be overwhelming formed in the process of socialization where the associations (White, Positive), (Black, Negative) are preponderant, if implicit. It was pointed out in [50] with respect to the United States that 'A critical aspect of the aversive racism framework is the conflict between Whites' denial of personal prejudice and the underlying unconscious negative feelings toward and belief about Blacks'. They argue that while the norms of society, and propagated by Whites are towards justice, fairness and racial equality, Whites have negative feelings towards Blacks of which they are unaware.

This unconscious bias is what the IAT attempts to measure. When participants are embodied as Black, bodily resonance through the novel data indicating that their own body can be Black, disrupts these implicit associations and reduces (but does not reverse) the implicit bias. This was discussed in [23,26] and has been formalized in a neural network model in [27]. This model also predicts the influence of self-esteem, since participants with high self-esteem would be more likely to have positive self-image associations, and thus be more likely to disrupt negative associations with Black as a result of embodiment. However, since almost all of our sample had normal to high self-esteem we are unable to provide further evidence with respect to this issue.

Our results, however, suggest that this analysis does not hold when there is negative affect. The relationship between emotion and cognition has been thoroughly studied in [51]. In particular, the authors discuss evidence of affective influence on implicit attitudes, that support our finding that in

the Negative situation new associations between Black and positive evaluations are less likely to form. It has been found in a series of experiments in [52,53] that manipulating the emotional state of subjects can influence their implicit bias. Moreover, the emotional state has to be commensurate with expectations regarding the outgroup: in their example, inducing 'disgust' would lead to bias against gay people but not Arabs, whereas inducing anger would induce bias against Arabs but not gay people in [52]. The Negative crowd behaved unpleasantly towards the participant, in accordance with what might be expected against that particular outgroup in a context of racial bias. Moreover, it has been suggested that negative affect can result in cognitive inhibition, affecting perception, attention, memory and learning [54], and therefore impair the formation and use of new implicit associations. We suggest that the negative situation therefore (i) prevented the formation of new associations that would favour Black, (ii) in itself the negative affect leads to greater bias following the findings in [52], and (iii) the negative response of the crowd while embodied as Black could only serve to reinforce the implicit associations between Black and negative attributes, even if at the conscious level the participant may not be biased.

This last point follows also from the neural network model [27] which is based on the notion that while doing the IAT people are more likely to select strongly supported associations. When in a dark-skinned body in a neutral or positive affective condition the resonance between self and the virtual body leads to a weakening of associations between Black and negative evaluation, relying on positive associations with self (for those with high esteem). Similarly being embodied as Black with hostile responses from the crowd, would help to reinforce the connection between Black and negative, notwithstanding any consciously held beliefs against racial bias. Indeed we found that there are no salient effects at all of any of the conditions on explicit bias measured through the ATB questionnaire.

A limitation of our study is that it involved only female participants for comparison with our previous studies. There is some evidence of differences in implicit and explicit racial bias between males and females [55], where it was found that explicit bias was systematically higher among men than women, but the implicit bias was systematically higher among women. Other studies using the IAT, in the context of the rubber hand illusion, have had both male and female participants and no differences were reported [22,40,42]. With respect to body ownership, [56] found no differences between males and females, even though both groups were embodied in a male virtual body. However, this issue is certainly worthy of further study. In [57] it was found that females with strong body ownership in a female virtual body would tolerate more changes to the shape of that body before they noticed these than males similarly embodied in a male virtual body. So it could be the case that there is some deeper differential response in the body ownership illusion between males and females. It would be important in future studies to be able to deal with this problem. A second limitation is that we were primarily concerned with implicit bias. Although we included a measure of explicit bias, it should be noted that the scores indicate low explicit bias. This may be because the study was carried out in Spain where historically there is not the same deep-seated bias of White against Black as there may be in other countries. A similar cross-cultural study would be useful to investigate this. A third limitation is that the crowd members were all of 'White' appearance. Results may have been different if the crowd itself had been heterogeneous in this regard. However, the point of our set-up was to see whether different affective states influence changes in implicit bias, so had the negative response been less for a more heterogeneous crowd this would have not fulfilled the objective of our study. Finally, we did not attempt to measure empathy itself, since this was not our focus. It would be interesting to explore possible relationships between implicit bias, explicit bias and measures of empathy.

Returning to our opening remarks, our results may seem to undermine the popular idea that VR is an 'empathy machine'. What is meant by this is that it is possible to give people immersive first-hand negative experiences of aspects of the life of people from social groups, countries, races, or situations that are quite different from their own. The idea is that this exposure would then increase their empathy towards those people. A well-known example is the United Nations' virtual reality production 'Clouds Over Sidra',[1] by Chris Milk and Gabo Arora which depicted life in a Syrian refugee camp in Jordan, as narrated by a 12-year-old girl Sidra. A simulation of being in a Guantánamo-like prison cell being interrogated under conditions of stress as reported in [58], as well as several other examples by de la Peña including hunger lines,[2] violence against women[3] and prisoner isolation.[4] The goal of such productions is the same—to use VR as a method to enhance

---

[1] http://unvr.sdgactioncampaign.org/cloudsoversidra/#.XbHKti-Q1UM.

[2] https://emblematicgroup.com/experiences/hunger-in-la/.

[3] https://emblematicgroup.com/experiences/kiya/.

[4] https://emblematicgroup.com/experiences/solitary-confinement/.

empathy towards others. For example, the authors of [1], writing about the 'ultimate empathy machine', say that 'The promise … is to push beyond film and invite viewers to immerse themselves in different lives and worlds', identifying 150 examples published in the period 2012–2018.

Our own examples include domestic violence offenders being embodied as women subject to a violent verbal and physically intimidating assault by a (virtual) man [59,60], and mothers being embodied as children interacting with a virtual mother [61]. This was found to evoke empathy towards a child's needs.

However, none of these examples are about bias in the sense of a set of automatic associations between examples that elicit thoughts of a group (e.g. a face of a member of that group) and negative evaluations. The IAT essentially samples such associations. The issue with domestic violence offenders is not that they are 'biased' against women in the normal sense that a person from one race might be biased against people from another race. The offenders treat women (or at least their partners) badly—often claiming that their own violent behaviour is 'normal' [59]. There are also strong issues in relation to power relationships within couples where domestic violence occurs, which is nothing to do with 'bias' in the sense of racial discrimination, for example, [62,63].

From this discussion, we cannot compare the results of the present study with these 'empathy machine' examples, since they are not measuring the same type of response. However, our results do offer caution about the idea of using VR to foster empathy under situations with a negative valence. As pointed out by [50] there is a disjunct between positive explicit attitudes (in their example, of Whites towards Blacks in the US) and implicit attitudes (actual behaviours based on the unconscious bias). In fact, in both the study of [20] and the present experiment there was no evidence of explicit racial bias before or after the experience, but there were changes in IAT, and the IAT has been found to have better predictive power for social interaction than explicit measures [64]. So while it is possible that VR might be used to improve explicit attitudes towards an outgroup, it does not follow that it also improves implicit attitudes of bias and may actually make things worse when the situation portrayed is a negative one.

It is assumed that the successful enhancement of empathy towards a group will elicit a positive change in behaviour towards that group. However, this is not clear. For example, the authors of [65] in their discussion of [66] argue that empathy at best results in superficial helping, see also [67]. When we see a homeless person asking for money in the street, we might have empathy and give them some coins, but do we take serious action such as spending time to find accommodation for them? Paul Bloom [68] provides substantial evidence across a number of studies that empathy does not necessarily result in helping action, but argues instead for rational compassion. In this paper, we do not take a stand on this issue, but we do suggest that the concept of empathy is not needed to explain the types of results we have found in this, and other related papers.

Ethics. The experiment was approved by the Comissió Bioètica of Universitat de Barcelona with approval number IRB00003099. Participants gave informed written consent prior to the start of the experiment.

Data accessibility. All data and the programs for analysis are also available on https://www.kaggle.com/melslater/racial-bias-data-and-program.

Authors' contributions. M.S., D.B., M.B.-O., S.N. and S.S. designed the research. D.B. and S.N. carried out the research and collected and prepared the data. A.B. designed and programmed the scenarios. M.S. obtained the funding and analysed the data. M.S., D.B. and S.N. wrote the first draft of the paper and all authors contributed.

Competing interests. The authors declare no competing interests.

Funding. This research was funded under the European Research Council (ERC) Advanced Grant, Moments in Time in Immersive Virtual Environments (MoTIVE) (no. 742989).

Acknowledgements. We thank Dr Ramon Oliva who helped with the programming.

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
