## [Reviewer comments · Royal Society Open Science]

Review History

RSOS-200564.R0 (Original submission)

Review form: Reviewer 1

Is the manuscript scientifically sound in its present form?

Yes

Are the interpretations and conclusions justified by the results?

Yes

Is the language acceptable?

Yes

Do you have any ethical concerns with this paper?

No

Have you any concerns about statistical analyses in this paper?

No

Recommendation?

Accept with minor revision (please list in comments)

Comments to the Author(s)

The aim of the study was to investigate how social factors like possible threat from a negative crowd. The study is original and interesting. It challenges overly general interpretations of the effects that can be achieved with body ownership manipulations. More specifically, the study underscores the need to investigate carefully the respective contributions from bottom up and top down factors in modulation of body ownership effects. The study adds a very valuable new chapter to the literature on the modalities and the impact of embodiment manipulations.

I'd like to point out the high quality of data analysis and statistical analyses.

I have a few minor comments and suggestions.

1. Why was a between groups rather than a within groups design used?
2. Motivate why only female participants were used?
3. The argument based on the notion of body ownership is p11 line7-8 cannot be understood in the very succinct statement here. The bodyguard concept is a philosophical one and may not have a specific meaning outside conceptual discussions. Better leave that out or spell out in more detail.
4. Top down and bottom up are broad and often vague notions. They can best be anchored to some specific literature on perpetual vs. post perceptual effects for example.

A point of more discussion and alternative explanations:

One might challenge the interpretation in the paper for the finding that participants in a white body in a negative social situation score lower on embodiment than when embodied in the same situation as a black body. The authors' interpretation is that the black embodied person expects the negative reaction as she is assuming that the crowd is in any case negative about the black person and therefore no retreat from the embodiment is prompted. Furthermore, if this is the correct interpretation, this can be seen as an increase in racial bias. When participants are in a black body and the crowd is negative, this negativity is what is expected, based on the participants' notion that the crowd reacts to my black body because of racism. Thus the participants presumably attribute implicit racism to the crowd.

Might it not be the case that the effect of a negative crowd is experienced as an assault on the psychological integrity of the real participant and by extension, on the black embodiment? Rather than an intact real participant a self-distancing from the black embodiment? Reduced black body ownership in the negative crowd condition reflects processes in the participant rather than black embodiment. Indeed, reduced body ownership was also seen in the white skin/negative crowd condition. In this respect, the explanation of the effects in the black embodiment conditions could be still be articulated more clearly. The authors might have a look at Watson & de Gelder SciRep 2017 on white and black body emotion perception.

As it is essential to understand the psychological and neural basis of embodiment.

A concluding section spelling out the implications of the socio-affective relativity of embodiment would be welcome as this finding projects a change of perspective.

Review form: Reviewer 2

Is the manuscript scientifically sound in its present form?

No

Are the interpretations and conclusions justified by the results?

No

Is the language acceptable?

No

Do you have any ethical concerns with this paper?

No

Have you any concerns about statistical analyses in this paper?

Yes

Recommendation?

Reject

Comments to the Author(s)

This study examined the effects of (1) embodying a light vs dark-skinned avatar and (2) the social context of a negative vs neutral vs positive crowd, on explicit and implicit racial bias. The authors measured the strength of the ownership, agency and presence illusions through the administration of questionnaires, as well as the pre- and post-experiment explicit and implicit racial biases. The authors conclude that the social context may influence the feeling of ownership as well as the implicit racial bias.

This study addresses an intriguing topic, and this manuscript is potentially very interesting. However, I find the presentation of the data and the statistical analyses overly complex and inappropriate, and the manuscript poorly written. The study consists of one behavioral experiment with a simple design, still there are 3 main figures, 1 main tables, and 23 pages of supplementary material with 14 supplementary tables and 5 supplementary figures (!). This manuscript would benefit hugely from employing a simpler statistical approach, selecting only the most relevant results to present, and cutting down the word count in the discussion. Below are some of my major points.

- The experimental design, a 2x3 factorial design with the factors avatar skin color (black, white) and crowd (negative, neutral, positive), is simple and easy to understand. Visual inspection of the data (Fig S1-S2) also reveals that there was a strong ownership, agency and presence illusions in all conditions. There also seems to be interesting post vs pre effects for the IAT scores (Fig S5). The vast majority of studies on full-body and limb-ownership illusions analyze this kind data using a 2x3 ANOVA and pairwise t-tests, or some nonparametric variant. However, the results section presented here is unnecessarily complex and difficult to follow, and focuses on the details of a Bayesian model (which I don't understand the rationale for using for this kind of data) rather than interpretations of the results. I think it would do this paper a great favor of "dumbing down" the statistical analysis and presentation of the results, to make it readable and accessible to a general reader.

- The key analysis of the IAT scores is the post-versus-pre comparison across groups. The authors state that this comparison is invalid because the preIAT scores differs across groups. However, their assertion is unsound, because looking at the post-versus-pre change in IAT takes into account baseline differences.

- Only one of the three questionnaire items defined as indicating disownership, are actually somewhat pinpointing the feeling of disownership ("I felt as if the virtual avatar was not me."). The two other statements, "I felt as if I had two looking bodies" and "I felt that my virtual body resembled my own (real) body in terms of shape, skin tone, or other visual features", are not capturing the feeling of disownership. These two statements as usually used as control statements, to control for suggestibility and task compliance, and should be analyzed separately.

- Main figures should show the data in its most intuitive form. To me, it makes much more sense to present the actual questionnaire rating per condition (Fig S1) instead of Fig 2, and, instead of Fig 3, the pre-minus-post IAT should be shown.

- Throughout the paper, unintelligible variable names are used (e.g. “ybodyown1” referring to ownership, and “ybodyown2” referring to disownership). Using the same variable names in a manuscript as coded in a statistical software is not helpful to the reader. Please change these to names that makes sense and are easy to grasp.

- The discussion is too long. Please make it more to the point and, in each paragraph, relate to your own findings.

- page 9, line 43: preIAT and postIAT should be preATB and postATB, I assume.

- table 1: the table is duplicated.

Review form: Reviewer 3

Is the manuscript scientifically sound in its present form?

Yes

Are the interpretations and conclusions justified by the results?

No

Is the language acceptable?

Yes

Do you have any ethical concerns with this paper?

Yes

Have you any concerns about statistical analyses in this paper?

No

Recommendation?

Major revision is needed (please make suggestions in comments)

Comments to the Author(s)

In this ms, authors implemented a between subject design (Body: Black-White) and Crowd (Negative- Positive - Neutral) to investigate the effect of social context and type of embodied avatar of the reduction of racial bias.

It is a timing research questions, but I have several concerns. First, but I don't know if it is related to the Journal guidelines, I suggest to better organize the structure of the paper to improve its readability. For example, the amount of information in Supplementary Material or a proper description of participants is extremely confusing.

Here my suggestions:

Introduction

1) I suggest explaining more in detail why usually participants report the same level of ownership over different bodies. I mean, I think that the Introduction could be enriched by a more detailed explanation of the multisensory basis of body ownership and the fundamental constraints on body ownership illusions (Humanoid shape rule).

2) The Introduction should be enriched by literature supporting the hypothesis that social context is able to modulate body ownership. Moreover, authors should clearly state their predictions.

Methods

- 3) Did authors asked if participants have close contact with black people?
- 4) Why this study included only females? This seriously limits the generalizability of the results, and authors should clearly explain their choice and its impact in Discussion
- 5) Response Variables: I found this section quite confusing since it includes both a description of the questionnaires/ tests used, and results obtained from their administration. Could authors please insert a section for describing questionnaires/ tests and their psychometric proprieties and then report results in proper sections?
- 6) Why in the Abstract and in some parts of the Discussion, authors introduced the idea that the avatar is "skinny"? Is it relevant? Did authors have inclusion/exclusion criteria for BMI for their participants to control for this variable? Please, report the BMI of the participants, if yes.
- 7) PCA the factor loading of "notme" on the second factor (0.189) seems very low to justify its inclusion. Could authors statistically checked the difference between "body ownership" and "body disownership" to test if the illusion was achieved?
- 8) I don't understand the statistical analyses used to verify (manipulation check) that the social context was perceived as authors experimentally manipulated it. I understood that they did a PCA on scores from the second questionnaire, and then I did understand how they statistically compared the results obtained. Some statement in Supplementary Material, such as "It is clear that the Negative crowd elicited negative participant responses compared to the Neutral and Positive conditions." are not clear. Could authors please explain it?
- 9) Is there a significant difference in the score in IAT at baseline (" It is clear that by chance the mean preIAT is higher for (White, Negative) compared to the other conditions")? I understand that authors used this measure as covariate, but they should clarify and discuss it.
- 10) I understood the Self-esteem had no effect. Could authors re-run the analysis without this variable?
- 11) In general, could authors please avoid discussing results in this Discussion? Otherwise it is quite confusing to read them.

Discussion

- 12) I found Discussion extremely confusing. Some of information provided here should be introduced early in the Introduction to give the study the right motivation. Moreover, there are same repetition (the entire previous paragraph about previous findings with RHI and bodily illusions with black bodies).
 - 13) Most importantly, it fails to explain the results found. Authors should focus their efforts in explain why they found a 1) "a strong body ownership for (Black, Negative), and similar levels for all other conditions except for (White, Negative)" and 2) Overall, (White, Neutral) results in an increase in mean IAT, (Black, Neutral) and (Black, Positive) in a decrease, and (Black, Negative) in an increase.?
- One last thing. In my opinion, statement like this is "as if there is an implicit expectation that 'bad things may happen' to a Black person but not to a White person, and this is reflected in a reduction of body ownership" are highly speculative.

Minor

- The first section of Material and Methods should be Participants, not Experimental Setup. Moreover, information about the Rosenberg self-esteem scale [24] should go in the section called Response Variables.
- Table S4 "Factor Analysis for responses to the crowd using all the scores of Table S1." I think that this is a typo, please check since this is not related to Table S1
- Some statement in Supplementary Material, such as "It is clear that the Negative crowd elicited negative participant responses compared to the Neutral and Positive conditions." or "All scores are high" are not particularly appropriate.

Decision letter (RSOS-200564.R0)

Dear Dr Slater:

Manuscript ID RSOS-200564 entitled "Virtual body ownership and its consequences for implicit racial bias are dependent on social context" which you submitted to Royal Society Open Science, has been reviewed. The comments from reviewers are included at the bottom of this letter.

In view of the criticisms of the reviewers, the manuscript has been rejected in its current form. However, a new manuscript may be submitted which takes into consideration these comments.

Please note that resubmitting your manuscript does not guarantee eventual acceptance, and that your resubmission will be subject to peer review before a decision is made.

Your resubmitted manuscript should be submitted by 29-Dec-2020. If you are unable to submit by this date please contact the Editorial Office.

Kind regards,
Andrew Dunn
Senior Publishing Editor
Royal Society Open Science
openscience@royalsociety.org

on behalf of Dr Giorgia Silani (Associate Editor)
openscience@royalsociety.org

Associate Editor Comments to Author (Dr Giorgia Silani):

We have now received the reviews of your manuscript referenced above. While the reviewers find interest in your data, they have also raised a number of serious concerns. Indeed, in order to have impact, we think that it should be more focused and concise. Furthermore, a better description of the data and the statistical analyses is recommended, given it seems to be overly complex. These concerns are outlined in their reviews which have been included below. Given the required revisions are consistent, we believe the "reject and allow to resubmit" option to be more appropriate, given it will give you more time to address reviewers' concerns.

Reviewers' Comments to Author:

Reviewer: 1

Comments to the Author(s)

The aim of the study was to investigate how social factors like possible threat from a negative

The study is original and interesting. It challenges overly general interpretations of the effects that can be achieved with body ownership manipulations. More specifically, the study underscores the need to investigate carefully the respective contributions from bottom up and top down factors in modulation of body ownership effects. The study adds a very valuable new chapter to the literature on the modalities and the impact of embodiment manipulations.

I'd like to point out the high quality of data analysis and statistical analyses.

I have a few minor comments and suggestions.

1. Why was a between groups rather than a within groups design used?
2. Motivate why only female participants were used?
3. The argument based on the notion of body ownership is p11 line7-8 cannot be understood in the very succinct statement here. The bodyguard concept is a philosophical one and may not have a specific meaning outside conceptual discussions. Better leave that out or spell out in more detail.
4. Top down and bottom up are broad and often vague notions. They can best be anchored to some specific literature on perpetual vs. post perceptual effects for example.

A point of more discussion and alternative explanations:

One might challenge the interpretation in the paper for the finding that participants in a white body in a negative social situation score lower on embodiment than when embodied in the same situation as a black body. The authors' interpretation is that the black embodied person expects the negative reaction as she is assuming that the crowd is in any case negative about the black person and therefore no retreat from the embodiment is prompted. Furthermore, if this is the correct interpretation, this can be seen as an increase in racial bias. When participants are in a black body and the crowd is negative, this negativity is what is expected, based on the participants' notion that the crowd reacts to my black body because of racism. Thus the participants presumably attribute implicit racism to the crowd.

Might it not be the case that the effect of a negative crowd is experienced as an assault on the psychological integrity of the real participant and by extension, on the black embodiment? Rather than an intact real participant a self-distancing from the black embodiment? Reduced black body ownership in the negative crowd condition reflects processes in the participant rather than black embodiment. Indeed, reduced body ownership was also seen in the white skin/negative crowd condition. In this respect, the explanation of the effects in the black embodiment conditions could be still be articulated more clearly. The authors might have a look at Watson & de Gelder SciRep 2017 on white and black body emotion perception.

As it is essential to understand the psychological and neural basis of embodiment.

A concluding section spelling out the implications of the socio-affective relativity of embodiment would be welcome as this finding projects a change of perspective.

Reviewer: 2

Comments to the Author(s)

This study examined the effects of (1) embodying a light vs dark-skinned avatar and (2) the social context of a negative vs neutral vs positive crowd, on explicit and implicit racial bias. The authors measured the strength of the ownership, agency and presence illusions through the administration of questionnaires, as well as the pre- and post-experiment explicit and implicit racial biases. The authors conclude that the social context may influence the feeling of ownership as well as the implicit racial bias.

This study addresses an intriguing topic, and this manuscript is potentially very interesting. However, I find the presentation of the data and the statistical analyses overly complex and inappropriate, and the manuscript poorly written. The study consists of one behavioral experiment with a simple design, still there are 3 main figures, 1 main table, and 23 pages of supplementary material with 14 supplementary tables and 5 supplementary figures (!). This

manuscript would benefit hugely from employing a simpler statistical approach, selecting only the most relevant results to present, and cutting down the word count in the discussion. Below are some of my major points.

- The experimental design, a 2x3 factorial design with the factors avatar skin color (black, white) and crowd (negative, neutral, positive), is simple and easy to understand. Visual inspection of the data (Fig S1-S2) also reveals that there was a strong ownership, agency and presence illusions in all conditions. There also seems to be interesting post vs pre effects for the IAT scores (Fig S5). The vast majority of studies on full-body and limb-ownership illusions analyze this kind data using a 2x3 ANOVA and pairwise t-tests, or some nonparametric variant. However, the results section presented here is unnecessarily complex and difficult to follow, and focuses on the details of a Bayesian model (which I don't understand the rationale for using for this kind of data) rather than interpretations of the results. I think it would do this paper a great favor of "dumbing down" the statistical analysis and presentation of the results, to make it readable and accessible to a general reader.
- The key analysis of the IAT scores is the post-versus-pre comparison across groups. The authors state that this comparison is invalid because the preIAT scores differs across groups. However, their assertion is unsound, because looking at the post-versus-pre change in IAT takes into account baseline differences.
- Only one of the three questionnaire items defined as indicating disownership, are actually somewhat pinpointing the feeling of disownership ("I felt as if the virtual avatar was not me."). The two other statements, "I felt as if I had two looking bodies" and "I felt that my virtual body resembled my own (real) body in terms of shape, skin tone, or other visual features", are not capturing the feeling of disownership. These two statements as usually used as control statements, to control for suggestibility and task compliance, and should be analyzed separately.
- Main figures should show the data in its most intuitive form. To me, it makes much more sense to present the actual questionnaire rating per condition (Fig S1) instead of Fig 2, and, instead of Fig 3, the pre-minus-post IAT should be shown.
- Throughout the paper, unintelligible variable names are used (e.g. "ybodyown1" referring to ownership, and "ybodyown2" referring to disownership). Using the same variable names in a manuscript as coded in a statistical software is not helpful to the reader. Please change these to names that makes sense and are easy to grasp.
- The discussion is too long. Please make it more to the point and, in each paragraph, relate to your own findings.
- page 9, line 43: preIAT and postIAT should be preATB and postATB, I assume.
- table 1: the table is duplicated.

Reviewer: 3

Comments to the Author(s)

In this ms, authors implemented a between subject design (Body: Black-White) and Crowd (Negative- Positive - Neutral) to investigate the effect of social context and type of embodied avatar of the reduction of racial bias.

It is a timing research questions, but I have several concerns. First, but I don't know if it is related to the Journal guidelines, I suggest to better organize the structure of the paper to improve its readability. For example, the amount of information in Supplementary Material or a proper description of participants is extremely confusing.

Here my suggestions:

Introduction

- 1) I suggest explaining more in detail why usually participants report the same level of ownership over different bodies. I mean, I think that the Introduction could be enriched by a more detailed explanation of the multisensory basis of body ownership and the fundamental constraints on body ownership illusions (Humanoid shape rule).
- 2) The Introduction should be enriched by literature supporting the hypothesis that social context is able to modulate body ownership. Moreover, authors should clearly state their predictions.

Methods

- 3) Did authors asked if participants have close contact with black people?
- 4) Why this study included only females? This seriously limits the generalizability of the results, and authors should clearly explain their choice and its impact in Discussion
- 5) Response Variables: I found this section quite confusing since it includes both a description of the questionnaires/ tests used, and results obtained from their administration. Could authors please insert a section for describing questionnaires/ tests and their psychometric proprieties and then report results in proper sections?
- 6) Why in the Abstract and in some parts of the Discussion, authors introduced the idea that the avatar is "skinny"? Is it relevant? Did authors have inclusion/exclusion criteria for BMI for their participants to control for this variable? Please, report the BMI of the participants, if yes.
- 7) PCA the factor loading of "notme" on the second factor (0.189) seems very low to justify its inclusion. Could authors statistically checked the difference between "body ownership" and "body disownership" to test if the illusion was achieved?
- 8) I don't understand the statistical analyses used to verify (manipulation check) that the social context was perceived as authors experimentally manipulated it. I understood that they did a PCA on scores from the second questionnaire, and then I did understand how they statistically compared the results obtained. Some statement in Supplementary Material, such as "It is clear that the Negative crowd elicited negative participant responses compared to the Neutral and Positive conditions." are not clear. Could authors please explain it?
- 9) Is there a significant difference in the score in IAT at baseline (" It is clear that by chance the mean preIAT is higher for (White, Negative) compared to the other conditions")? I understand that authors used this measure as covariate, but they should clarify and discuss it.
- 10) I understood the Self-esteem had no effect. Could authors re-run the analysis without this variable?
- 11) In general, could authors please avoid discussing results in this Discussion? Otherwise it is quite confusing to read them.

Discussion

- 12) I found Discussion extremely confusing. Some of information provided here should be introduced early in the Introduction to give the study the right motivation. Moreover, there are some repetitions (the entire previous paragraph about previous findings with RHI and bodily illusions with black bodies).
 - 13) Most importantly, it fails to explain the results found. Authors should focus their efforts in explaining why they found a 1) "a strong body ownership for (Black, Negative), and similar levels for all other conditions except for (White, Negative)" and 2) Overall, (White, Neutral) results in an increase in mean IAT, (Black, Neutral) and (Black, Positive) in a decrease, and (Black, Negative) in an increase.?
- One last thing. In my opinion, statement like this is "as if there is an implicit expectation that 'bad things may happen' to a Black person but not to a White person, and this is reflected in a reduction of body ownership" are highly speculative.

Minor

- The first section of Material and Methods should be Participants, not Experimental Setup. Moreover, information about the Rosenberg self-esteem scale [24] should go in the section called Response Variables.

- Table S4 “Factor Analysis for responses to the crowd using all the scores of Table S1.” I think that this is a typo, please check since this is not related to TableS1

- Some statement in Supplementary Material, such as “It is clear that the Negative crowd elicited negative participant responses compared to the Neutral and Positive conditions.” or “All scores are high” are not particularly appropriate.

Author's Response to Decision Letter for (RSOS-200564.R0)

See Appendix A.

RSOS-201848.R0

Review form: Reviewer 1

Is the manuscript scientifically sound in its present form?

Yes

Are the interpretations and conclusions justified by the results?

Yes

Is the language acceptable?

Yes

Do you have any ethical concerns with this paper?

No

Have you any concerns about statistical analyses in this paper?

No

Recommendation?

Accept as is

Comments to the Author(s)

The authors have done a very thorough job in addressing the issues raised by the review. I have no further comments.

Review form: Reviewer 2

Is the manuscript scientifically sound in its present form?

No

Are the interpretations and conclusions justified by the results?

Yes

Is the language acceptable?

No

Do you have any ethical concerns with this paper?

No

Have you any concerns about statistical analyses in this paper?

Yes

Recommendation?

Reject

Comments to the Author(s)

I am sorry but I do not feel that my main concerns have been addressed satisfactorily. To me, the statistical approach remains overly complex for such a simple experiment. The topic is indeed interesting, but the manuscript overall is confusing and extremely hard to read, with way too long introduction and discussion sections, and too much supplementary material. I do not feel that the manuscript reaches the necessary quality standard for publication in this journal.

Review form: Reviewer 3

Is the manuscript scientifically sound in its present form?

Yes

Are the interpretations and conclusions justified by the results?

Yes

Is the language acceptable?

Yes

Do you have any ethical concerns with this paper?

No

Have you any concerns about statistical analyses in this paper?

No

Recommendation?

Accept as is

Comments to the Author(s)

The authors have satisfactorily responded to all my questions.

Decision letter (RSOS-201848.R0)

Dear Dr Slater,

I am pleased to inform you that your manuscript entitled "Virtual body ownership and its consequences for implicit racial bias are dependent on social context" is now accepted for publication in Royal Society Open Science.

Royal Society Open Science operates under a continuous publication model. Your article will be published as soon as it is ready for publication, and this will be the final version of the paper. As such, it can be cited immediately by other researchers. As the issue version of your paper will be the only version to be published I would advise you to check your proofs thoroughly as changes cannot be made once the paper is published.

Articles are normally press released. For this to be effective we set an embargo on news coverage corresponding to the publication date of the article. We request that news media and the authors do not publish stories ahead of this embargo (when final version of the article is available). Please see the Royal Society Publishing guidance on how you may share your accepted author manuscript at <https://royalsociety.org/journals/ethics-policies/media-embargo/>.

on behalf of Dr Giorgia Silani (Associate Editor)
openscience@royalsociety.org

Associate Editor Comments to Author (Dr Giorgia Silani):

Associate Editor

Comments to the Author:

We are very pleased to announce that your paper has been accepted for publication! Two reviewers are very positive and agreed that the changes provided are satisfactory. One reviewer still think that the statistical analysis is overly complex. In spite of this criticism, we believe that the quality of the paper has reached the necessary standard for publication in the journal.

Reviewer comments to Author:

Reviewer: 1

Comments to the Author(s)

The authors have done a very thorough job in addressing the issues raised by the review.

I have no further comments.

Reviewer: 2

Comments to the Author(s)

I am sorry but I do not feel that my main concerns have been addressed satisfactorily. To me, the statistical approach remains overly complex for such a simple experiment. The topic is indeed interesting, but the manuscript overall is confusing and extremely hard to read, with way too long introduction and discussion sections, and too much supplementary material. I do not feel that the manuscript reaches the necessary quality standard for publication in this journal.

Reviewer: 3

Comments to the Author(s)

The authors have satisfactorily responded to all my questions.

Appendix A

We would like to thank all the reviewers for the very helpful feedback and the constructive comments.

Reviewer: 1

The aim of the study was to investigate how social factors like possible threat from a negative The study is original and interesting. It challenges overly general interpretations of the effects that can be achieved with body ownership manipulations. More specifically, the study underscores the need to investigate carefully the respective contributions from bottom up and top down factors in modulation of body ownership effects. The study adds a very valuable new chapter to the literature on the modalities and the impact of embodiment manipulations.

I d like to point out the high quality of data analysis and statistic analyses.

I have a few minor comments and suggestions.

1. Why was a between groups rather than a within groups design used?

One reason for following a between-groups design was to avoid exposing participants to the IAT test multiple times. In general, the test shows a relative weak test-retest reliability (test overall at 0.6), while it has also been argued that scores can vary between multiple administrations with repeated exposure known to decrease the magnitude of the effect [1]. This was also why we decided to record baseline IAT scores approximately a week prior to participants' VR experience, following earlier examples [2-4]. An additional control to that was the counterbalancing of the order of the combined blocks of the IAT between participants as proposed in [5].

Equally important was to avoid the possibility of demand characteristics. If participants experienced all three conditions then they could easily guess the purpose of the experiment.

2. Motivate why only female participants were used?

As we point out in our Discussion, we agree this was a limitation of the study that needs to be addressed in future research. However, this was done firstly for comparison with our previous studies [2, 3], where only female participants were recruited and also because there is some evidence of differences in implicit and explicit racial bias between males and females [6]. It was found that explicit bias was systematically higher amongst men than women, but implicit bias was systematically higher amongst women. Other studies in this domain have had both male and female participants and not reported differences, for example [7-9]. We do not expect important differences with males, however, we agree it would be best to have both. We have extended the discussion of this mentioning other important work [10] that suggests that there may be differences in body ownership illusions between males and females, and pointed out the need to address this in future studies.

3. The argument based on the notion of body ownership is p11 line7-8 cannot be understood in the very succinct statement here. The bodyguard concept is a philosophical one and may not have a specific meaning outside conceptual discussions. Better leave that out or spell out in more detail.

We assume that this refers to “This is consequent on the bodyguard hypothesis [43], where body ownership implies self-protection.” We agree with this comment and we have removed the reference.

4. Top down and bottom up are broad and often vague notions. They can best be anchored to some specific literature on perpetual vs. post perceptual effects for example.

We refer to these notions specifically within the body ownership illusions context, where previous work has focused on the perceptual rules that determine the rubber-hand illusion and similar ownership illusions. It has been established that such experiences rely on the manipulation of bottom-up multisensory signals (vision, touch, proprioception etc.) that contribute to the sense of bodily self. In addition to these, body ownership may also be influenced by top-down processes, such as our expectations of reality (e.g., a first-person perspective or an anatomically plausible position of the seen body, continuity between body parts etc.) and internal models of our own body appearance [11-13]. We refer to these in more detail with examples and also suggest based on our results that whereas bottom-up stimuli and top-down in relation to aspects of the surrogate body, and consistency between these two, are required for illusory body ownership, the affective social situation in which the embodiment takes place is also important. We have moved this part in the introduction following another reviewer’s recommendation and we return to this point in the discussion after the findings of reduced ownership in the (White, Negative) condition.

Additionally, of relevance here is also the Proteus Effect we refer to in the introduction, whereby when people are virtually embodied or represented online with a virtual body different to their own then they exhibit behaviours concomitant with attributes of that body. As we discuss similar results have been replicated in a number of studies over distinct bodies with various physical characteristics, including race as in this case.

A point of more discussion and alternative explanations:

One might challenge the interpretation in the paper for the finding that participants in a white body in a negative social situation score lower on embodiment than when embodied in the same situation as a black body. The authors' interpretation is that the black embodied person expects the negative reaction as she is assuming that the crowd is in any case negative about the black person and therefore no retreat from the embodiment is prompted. Furthermore, if this is the correct interpretation, this can be seen as an increase in racial bias.

When participants are in a black body and the crowd is negative, this negativity is what is expected, based on the participants' notion that the crowd reacts to my black body because of racism. Thus the participants presumably attribute implicit racism to the crowd.

Yes, this is an important point that we overlooked. We were thinking from the point of view of implicit expectations on the part of the participants, they experience what they would expect to experience. Of course this does imply that they attribute implicit racism to the crowd. We have made a comment about this.

Might it not be the case that the effect of a negative crowd is experienced as an assault on the psychological integrity of the real participant and by extension, on the black embodiment? Rather than an intact real participant a self-distancing from the black embodiment? Reduced black body ownership in the negative crowd condition reflects processes in the participant rather than black embodiment. Indeed, reduced body ownership was also seen in the white skin/negative crowd condition.

What happened though is that body ownership was not reduced in the (Black, Negative) condition, only in the (White, Negative) condition (there was an error in the original abstract). This is related to the point above – while the Negative behaviour of the crowd could be a predicted outcome in the Black embodiment condition (based on imputing implicit racism to the crowd), the Negative behaviour of the crowd would not be expected in the White condition. Our argument was that body ownership would be reduced in this condition as a way of distancing the self from the negative affect.

In this respect, the explanation of the effects in the black embodiment conditions could be still be articulated more clearly. The authors might have a look at Watson & de Gelder SciRep 2017 on white and black body emotion perception.

As it is essential to understand the psychological and neural basis of embodiment.

We thank the reviewer for bringing the paper of [14] to our attention. We now consider this paper in the Discussion.

A concluding section spelling out the implications of the socio-affective relativity of embodiment would be welcome as this finding projects a change of perspective.

We address some of the implications that arise as a function of this socio-affective relation of embodiment and the idea that VR can be used as ‘empathy machine’ through a change in perspective-taking in the Discussion. We argue that while it is possible that VR and embodiment techniques might be used to improve explicit attitudes towards an outgroup, it does not follow that it also improves implicit attitudes of bias, and may actually make things worse under negative affective situations. We consider some arguments from the literature whereby empathy at best results in superficial helping rather than actual action. Of course, we also point out that we cannot directly compare the results of the present study with previous examples addressing embodiment and perspective-taking under different socio-affective situations, since they are not measuring the same type of response.

Reviewer: 2

This study examined the effects of (1) embodying a light vs dark-skinned avatar and (2) the social context of a negative vs neutral vs positive crowd, on explicit and implicit racial bias. The authors measured the strength of the ownership, agency and presence illusions through the administration of questionnaires, as well as the pre- and post-experiment explicit and implicit racial biases. The authors conclude that the social context may influence the feeling of ownership as well as the implicit racial bias.

This study addresses an intriguing topic, and this manuscript is potentially very interesting.

However, I find the presentation of the data and the statistical analyses overly complex and inappropriate, and the manuscript poorly written. The study consists of one behavioral experiment with a simple design, still there are 3 main figures, 1 main table, and 23 pages of supplementary material with 14 supplementary tables and 5 supplementary figures (!). This manuscript would benefit hugely from employing a simpler statistical approach, selecting only the most relevant results to present, and cutting down the word count in the discussion.

We have dramatically reduced the supplementary information. This was due to deployment of a more concise (though equivalent) statistical model. Therefore the supplementary information now consists of further details on the programming implementation, more details on the IAT and ATB and evaluation of affective responses to the crowds.

Below are some of my major points.

- The experimental design, a 2x3 factorial design with the factors avatar skin color (black, white) and crowd (negative, neutral, positive), is simple and easy to understand. Visual inspection of the data (Fig S1-S2) also reveals that there was a strong ownership, agency and presence illusions in all conditions. There also seems to be interesting post vs pre effects for the IAT scores (Fig S5). The vast majority of studies on full-body and limb-ownership illusions analyze this kind of data using a 2x3 ANOVA and pairwise t-tests, or some nonparametric variant. However, the results section presented here is unnecessarily complex and difficult to follow, and focuses on the details of a Bayesian model (which I don't understand the rationale for using for this kind of data) rather than interpretations of the results. I think it would do this paper a great favor of "dumbing down" the statistical analysis and presentation of the results, to make it readable and accessible to a general reader.

In the revised paper we have greatly simplified the method. Instead of modelling the response via the parameters in the form of regression-like equations, we directly model the responses according to their theoretical means in the 2x3 factorial design. This substantially simplifies the presentation, and removes the necessity for a large amount of the supplementary information. All main results are now given in the manuscript.

We continue using a Bayesian method. Classical statistical methods, using null hypothesis significance testing, have a standard way to present results ($P < 0.05$). However, not only is this approach more and more discredited, it cannot satisfactorily deal with multiple tests, where overall control of ‘significance’ is lost, and ad-hoc methods are resorted to in order to overcome this. Bayesian methods do not suffer from this drawback – there is no ‘live’ or ‘die’ cut-off significance level, and the method produces one overall model that encapsulates all response variables simultaneously, so that findings about individual parameters are derived from the joint distribution of all parameters. Hence there is no question of the problem of ‘multiple comparisons’ and the concomitant problem with significance.

- The key analysis of the IAT scores is the post-versus-pre comparison across groups. The authors state that this comparison is invalid because the preIAT scores differs across groups. However, their assertion is unsound, because looking at the post-versus-pre change in IAT takes into account baseline differences.

Suppose we have the preIAT scores $preIAT_i$ and postIAT score $postIAT_i, i = 1, 2, \dots n$. Taking the response variable as the difference $postIAT_i - preIAT_i$ assumes that the statistical model is of the form:

$$postIAT_i = preIAT_i + rest\ of\ model \dots \tag{A}$$

This is a strong restriction on the relationship between pre- and postIAT, compared to the model

$$postIAT_i = \beta \times preIAT_i + rest\ of\ model \dots \tag{B}$$

In the case of our data, β is clearly not 1, it has 95% credible interval 0.17 to 0.57, the probability that $\beta < 1$ is 1.000. This is also clear from Figure 5C (new manuscript). Hence the second model would have better explanatory power than the first, the first model being clearly weaker. Of course sometimes the assumption that $\beta \approx 1$ will be justified, as for example in the case of the ATB scores, and therefore the differences could be used as the response. But that is not the case with these IAT data.

We have formally tested between the two models (A) and (B) above. This is described in the section “Goodness of fit and model comparisons”. It shows that (A) results in a clear reduction of an information criterion statistic compared with (B) (the mean reduction is 3 times its standard error).

Moreover, as can be seen from Figure 5C, and from the results in Table 2, the relationship between postIAT and preIAT is of the form:

$$\text{postIAT} = a + b \cdot \text{preIAT}.$$

Therefore if we take the difference:

$$\text{postIAT} - \text{preIAT} = a + (b-1) \cdot \text{preIAT}.$$

In the case that $b \neq 1$ the size of the difference depends on preIAT. In particular we have $b < 1$, so that greater values of preIAT magnify the difference. In the case of these data the preIAT level is, by chance, greater for (White, Negative) than all of the others (Figure 5A). Therefore looking at the differences (Figure 5B) it can be seen that it is the lowest value. This gives a biased view of the influence of the experimental factors, since preIAT is clearly influencing these differences. In other words the differences confound the influence of the experimental factors and the influence of preIAT. Hence our method of using preIAT as a covariate, which is also justified by the formal analysis of the difference between the two models, as described above.

- Only one of the three questionnaire items defined as indicating disownership, are actually somewhat pinpointing the feeling of disownership (“I felt as if the virtual avatar was not me.”). The two other statements, “I felt as if I had two looking bodies” and “I felt that my virtual body resembled my own (real) body in terms of shape, skin tone, or other visual features”, are not capturing the feeling of disownership. These two statements as usually used as control statements, to control for suggestibility and task compliance, and should be analyzed separately.

The reviewer is right. We carried out a Cronbach’s alpha analysis, and found indeed that the only internally consistent questionnaire responses are ‘mybody’, ‘mirror’ and ‘notme’. We have therefore based all the analysis solely on these three. Moreover, as we found before a factor analysis shows that the principle factor is of the form: mybody + mirror – notme, and therefore we have used a summative variable proportional to this in our analysis (consistent with Cronbach’s alpha). The illusion is shown not only by the higher values of mybody and mirror than notme, but also the Cronbach’s alpha and factor analysis finds that mybody and mirror have positive loadings and notme negative loadings, and the three loadings are almost equal in absolute value.

- Main figures should show the data in its most intuitive form. To me, it makes much more sense to present the actual questionnaire rating per condition (Fig S1) instead of Fig 2, and, instead of Fig 3, the pre-minus-post IAT should be shown.

We have included all the graphs as suggested in the main manuscript .

- Throughout the paper, unintelligible variable names are used (e.g. “ybodyown1” referring to ownership, and “ybodyown2” referring to disownership). Using the same variable names in a manuscript as coded in a statistical software is not helpful to the reader. Please change these to names that makes sense and are easy to grasp.

We agree – the name of the single ownership variable (derived from the factor analysis) is now yown (we use ‘y’ to follow the convention that it is a constructed response variable). However, the other variable names correspond well with their meaning (e.g., preIAT, postIAT).

- The discussion is too long. Please make it more to the point and, in each paragraph, relate to your own findings.

We have made some changes in the Discussion but unfortunately, we were unable to reduce the length substantially as we needed to discuss our findings in relation to the existing literature. Also, some further recommendations were suggested by other reviewers that we deemed important to include. We specifically address two points related to our results a) that body ownership may depend on the affective social situation depicted in the scenario and b) the change in implicit racial bias amongst White people embodied in a Black virtual body may also be influenced by the social situation depicted. Each argument is discussed in turn, while we conclude with addressing our opening remark as to how VR may or may not be used as an ‘empathy machine’ and consider some limitations of the study.

- page 9, line 43: preIAT and postIAT should be preATB and postATB, I assume.

- table 1: the table is duplicated.

These have been fixed.

Reviewer: 3

In this ms, authors implemented a between subject design (Body: Black-White) and Crowd (Negative- Positive – Neutral) to investigate the effect of social context and type of embodied avatar of the reduction of racial bias.

It is a timing research questions, but I have several concerns. First, but I don’t know if it is related to the Journal guidelines, I suggest to better organize the structure of the paper to improve its readability. For example, the amount of information in Supplementary Material or a proper description of participants is extremely confusing.

We have greatly reduced Supplementary Material. This has been possible without any loss due to a simpler formulation of the statistical model, where we directly model the means in the 2×3 design, rather than formulating the model with parameters as in a regression. The model is mathematically equivalent to the original, and has the same results.

We have moved all results to the Results section, except those pertaining to the evaluations of the crowd, since this is not part of our principle findings, but negative affect being generated by the negative crowd is a prerequisite for the main findings. In other words we are not attempting to generalise from sample to population regarding affective responses to the crowd, but only trying to show that there were the expected variations in affect. Hence we have left this in the Supplementary in order to not break the flow of the main paper. However, there is a section on responses to the crowd in the main manuscript where we have summarised the results and referred to the Supplementary. .

Here my suggestions:

Introduction

1) I suggest explaining more in detail why usually participants report the same level of ownership over different bodies. I mean, I think that the Introduction could be enriched by a more detailed explanation of the multisensory basis of body ownership and the fundamental constraints on body ownership illusions (Humanoid shape rule).

We refer to a detailed explanation of the multisensory nature of body ownership illusions in the discussion through specific examples, and further suggest based on our results that whereas bottom-up stimuli and top-down in relation to aspects of the surrogate body, and consistency between these two, are required for illusory body ownership, the social situation in which the embodiment takes place can also influence this.

We have moved the relevant section to the introduction instead following the reviewer's recommendation, and we return to this point in the discussion following the finding of reduced ownership after embodiment in (White, Negative).

2) The Introduction should be enriched by literature supporting the hypothesis that social context is able to modulate body ownership. Moreover, authors should clearly state their predictions.

The finding that body ownership may depend on the social context was an unexpected one and not part of our original hypotheses as previous literature on body ownership illusions over different bodies show no differences between black and white virtual bodies or rubber hands, child or adult surrogate bodies and so on. Thus, we deemed it appropriate to discuss the findings on body ownership and social context in more detail in the Discussion section. Nonetheless, we agree that our hypotheses were not clearly stated and we have made the appropriate changes in the introduction to reflect this.

Methods

3) Did authors asked if participants have close contact with black people?

We did not ask participants if they have close contact with black people, but they belonged to a homogeneous population with no (or probably very little) real life experience with black populations. This is the situation in Barcelona where the experiment was carried out.

4) Why this study included only females? This seriously limits the generalizability of the results, and authors should clearly explain their choice and its impact in Discussion

We discuss this in the limitations of our study, and we do agree this is a problem that needs to be addressed in future research. This was done firstly for comparison with our previous studies [2, 3], where only female participants were recruited and also because there is some evidence of differences in implicit and explicit racial bias between males and females [6]. It was found that explicit bias is systematically higher amongst men than women, but implicit bias is systematically higher amongst women. We have at the end of the Discussion referenced various other studies that included males and females where no differences were reported. In particular we have mentioned a new study on body ownership [10] that suggests that there may be some difference in ownership illusions between males and females, and that therefore this issue might be important and followed up in later work.

5) Response Variables: I found this section quite confusing since it includes both a description of the questionnaires/tests used, and results obtained from their administration. Could authors please insert a section for describing questionnaires/tests and their psychometric properties and then report results in proper sections?

We agree and this has been changed. In particular we have carried out a Cronbach's alpha analysis on the questionnaire scores concerned with body ownership.

6) Why in the Abstract and in some parts of the Discussion, authors introduced the idea that the avatar is "skinny"? Is it relevant? Did authors have inclusion/exclusion criteria for BMI for their participants to control for this variable? Please, report the BMI of the participants, if yes.

This comment is not related to our paper which has no discussion about 'skinny' avatars or BMI.

7) PCA the factor loading of "notme" on the second factor (0.189) seems very low to justify its inclusion. Could authors statistically checked the difference between "body ownership" and "body disownership" to test if the illusion was achieved?

The discussion of the body ownership questions has been thoroughly revised with the use of Cronbach's alpha, and the consequent derivation of a slightly different summative measure of ownership.

8) I don't understand the statistical analyses used to verify (manipulation check) that the social context was perceived as authors experimentally manipulated it. I understood that they did a PCA on scores from the second questionnaire, and then I did understand how they statistically compared the results obtained. Some statement in Supplementary Material, such as "It is clear that the Negative crowd elicited negative participant responses compared to the Neutral and Positive conditions." are not clear. Could authors please explain it?

We have included an analysis of the responses to the crowd, which supports the contention that the Negative crowd had negative affect compared to the others. This is described in detail in Supplementary Text S4, Figures S1-S4. We have left this in the Supplementary Material, in order to avoid breaking the flow in the main manuscript. The issue of the effect of the crowd is a pre-condition for the experiment, and not in itself a principle finding.

9) Is there a significant difference in the score in IAT at baseline ("It is clear that by chance the mean preIAT is higher for (White, Negative) compared to the other conditions")? I understand that authors used this measure as covariate, but they should clarify and discuss it.

In the results section we have added "(As an indicator of the strength of the difference, the Wilcoxon Rank Sum statistic is $z = -2.72$, $P = 0.0057$)".

10) I understood the Self-esteem had no effect. Could authors re-run the analysis without this variable?

Yes, this has been done. We have removed self-esteem from the analysis, and explained that it had no effect and why not.

11) In general, could authors please avoid discussing results in this Discussion? Otherwise it is quite confusing to read them.

We have corrected this. We have not included results in the Discussion, but we have, of course, discussed the results.

Discussion

12) I found Discussion extremely confusing. Some of information provided here should be introduced early in the Introduction to give the study the right motivation. Moreover, there are same repetition (the entire previous paragraph about previous findings with RHI and bodily illusions with black bodies).

In the Discussion, we specifically address two points related to our results a) that body ownership may depend on the affective social situation depicted in the scenario and b) the

change in implicit racial bias amongst white people embodied in a Black virtual body may also be influenced by the social situation depicted. Each argument is discussed in turn in relation to the existing literature. We conclude with addressing our opening remark as to how VR may or may not be used as an ‘empathy machine’ and consider some limitations of the study.

Some of the information introduced in the Discussion, such as ownership illusions in a social context, could not have been introduced earlier, because as explained above this was not part of our hypothesis but rather an unexpected findings in the analysis. Therefore, we considered it appropriate to discuss it following our results.

With respect to bodily illusions with black bodies as discussed specifically in the Discussion section, rather than this being a repetition of the previous literature, we wanted to point out that while in the previous studies reported above no differences were found between the subjective illusion of ownership between the conditions, we did find a difference here. This is relevant to our findings, where we show that the Neutral condition of our experiment provides a further replication of this, which is not the case for the Negative condition. We then go on addressing that point. We have restructured this paragraph to avoid any repetitions and focus on this point.

13) Most importantly, it fails to explain the results found. Authors should focus their efforts in explain why they found a 1) “a strong body ownership for (Black, Negative), and similar levels for all other conditions except for (White, Negative)” and 2) Overall, (White, Neutral) results in an increase in mean IAT, (Black, Neutral) and (Black, Positive) in a decrease, and (Black, Negative) in an increase.?

The fundamental explanations are (1) body ownership is reduced in (White, Negative) compared to the other conditions due to reduction of ownership in conditions of negative affect (we referred to other papers that have found this). This is our explanation based on the literature that in conditions of stress a dissociation from the body is one response. (2) the increase in IAT (implicit bias) in (Black, Negative) is due to the failure to make new associations under conditions of stress. We have modified the discussion.

One last thing. In my opinion, statement like this is “as if there is an implicit expectation that ‘bad things may happen’ to a Black person but not to a White person, and this is reflected in a reduction of body ownership” are highly speculative.

We agree and this has been removed.

Minor

- The first section of Material and Methods should be Participants, not Experimental Setup. Moreover, information about the Rosenberg self-esteem scale [24] should go in the section called Response Variables.

We agree and have changed this.

The Rosenberg self-esteem scale was not a response variable but was rather measured prior to the VR exposure to be used as a covariate because the theory developed in [15] suggests that participants in a Black virtual body are more likely to exhibit a reduction in racial bias the greater their self-esteem. In fact self-esteem had no effect at all, and we have discussed this, and explained that it had no effect because there was little variation in the sample, and most participants had normal to high self-esteem. We have taken it out of all subsequent analysis.

- Table S4 “Factor Analysis for responses to the crowd using all the scores of Table S1.” I think that this is a typo, please check since this is not related to TableS1

This has been corrected.

- Some statement in Supplementary Material, such as “It is clear that the Negative crowd elicited negative participant responses compared to the Neutral and Positive conditions.” or “All scores are high” are not particularly appropriate.

We have carried out a full analysis of the response to crowd data in Supplementary Text S4.

Regarding the data on presence (now section “Agency and Presence” in the main manuscript) there is really no analysis to do. Looking at the box plots it is the case that all the scores are very high (medians all 2 or 3 and all lower quartiles = 2, in the possible range of scores -3 to 3).

References

- 1 Greenwald, A. G., Nosek, B. A. 2001 Health of the Implicit Association Test at age 3. *Zeitschrift für Experimentelle Psychologie*. **48**, 85-93.
- 2 Peck, T. C., Seinfeld, S., Aglioti, S. M., Slater, M. 2013 Putting yourself in the skin of a black avatar reduces implicit racial bias. *Consciousness and cognition*. **22**, 779-787. (10.1016/j.concog.2013.04.016)
- 3 Banakou, D., PD, H., Slater, M. 2016 Virtual Embodiment of White People in a Black Virtual Body Leads to a Sustained Reduction in their Implicit Racial Bias. *Frontiers in Human Neuroscience*. **10:601**, (10.3389/fnhum.2016.00601)
- 4 Maister, L., Slater, M., Sanchez-Vives, M. V., Tsakiris, M. 2015 Changing bodies changes minds: owning another body affects social cognition. *Trends in Cognitive Sciences*. **19**, 6-12. (doi:10.1016/j.tics.2014.11.001)
- 5 Nosek, B. A., Greenwald, A. G., Banaji, M. R. 2007 The Implicit Association Test at age 7: A methodological and conceptual review. *Automatic processes in social thinking and behavior*. 265-292.
- 6 Ekehammar, B., Akrami, N., Araya, T. 2003 Gender differences in implicit prejudice. *Personality and Individual Differences*. **34**, 1509-1523.

- 7 Farmer, H., Maister, L., Tsakiris, M. 2014 Change my body, change my mind: the effects of illusory ownership of an outgroup hand on implicit attitudes toward that outgroup. *Frontiers in psychology*. **4**, 1016.
- 8 Lira, M., Egito, J. H., Dall'Agnol, P. A., Amodio, D. M., Gonçalves, Ó. F., Boggio, P. S. 2017 The influence of skin colour on the experience of ownership in the rubber hand illusion. *Scientific reports*. **7**, 15745.
- 9 Maister, L., Sebanz, N., Knoblich, G., Tsakiris, M. 2013 Experiencing ownership over a dark-skinned body reduces implicit racial bias. *Cognition*. **128**, 170-178. (10.1016/j.cognition.2013.04.002)
- 10 Jung, M., Kim, J., Kim, K. 2020 Measuring recognition of body changes over time: A human-computer interaction tool using dynamic morphing and body ownership illusion. *PLoS one*. **15**, e0239322.
- 11 Tsakiris, M. 2010 My body in the brain: a neurocognitive model of body-ownership. *Neuropsychologia*. **48**, 703-712.
- 12 Kalckert, A., Ehrsson, H. H. 2012 Moving a Rubber Hand that Feels Like Your Own: A Dissociation of Ownership and Agency. *Frontiers in human neuroscience*. **6**, (10.3389/fnhum.2012.00040)
- 13 Tsakiris, M., Haggard, P. 2005 The Rubber Hand Illusion Revisited : Visuotactile Integration and Self-Attribution. **31**, 80 -91. (10.1037/0096-1523.31.1.80)
- 14 Watson, R., De Gelder, B. 2017 How white and black bodies are perceived depends on what emotion is expressed. *Scientific reports*. **7**, 1-12.
- 15 Bedder, R. L., Bush, D., Banakou, D., Peck, T., Slater, M., Burgess, N. 2019 A mechanistic account of bodily resonance and implicit bias. *Cognition*. **184**, 1-10.